# Local Constrained Bayesian Optimization

**Jingzhe Jing** [1] [2]   **Zheyi Fan** [1] [2] [†]   **Szu Hui Ng** [3]   **Qingpei Hu** [1] [2] [†]

## Abstract

Bayesian optimization (BO) for high-dimensional constrained problems remains a significant challenge due to the curse of dimensionality. We propose **L**ocal **C**onstrained **B**ayesian **O**ptimization (LCBO), a novel framework tailored for such settings. Unlike trust-region methods that are prone to premature shrinking when confronting tight or complex constraints, LCBO leverages the differentiable landscape of constraint-penalized surrogates to alternate between rapid local descent and uncertainty-driven exploration. Theoretically, we prove that LCBO achieves a convergence rate for the Karush-Kuhn-Tucker (KKT) residual that depends polynomially on the dimension $d$ for common kernels under mild assumptions, offering a rigorous alternative to global BO where regret bounds typically scale exponentially. Extensive evaluations on high-dimensional benchmarks (up to 100D) demonstrate that LCBO consistently outperforms state-of-the-art baselines.

## 1. Introduction

Bayesian Optimization (BO) has established itself as a powerful paradigm for the global optimization of black-box functions that are expensive to evaluate (Shahriari et al., 2015). While standard BO focuses on unconstrained objectives, many real-world applications inherently involve unknown and expensive-to-evaluate constraints. Examples span various domains: in machine learning, hyperparameter tuning often requires optimizing validation accuracy subject to constraints on training time or memory usage (Snoek

---
[1] State Key Laboratory of Mathematical Sciences, Academy of Mathematics and Systems Science, Chinese Academy of Sciences, Beijing, China [2] School of Mathematical Sciences, University of Chinese Academy of Sciences, Beijing, China [3] Department of Industrial Systems Engineering & Management, National University of Singapore, Singapore. Correspondence to: Zheyi Fan <fanzheyi@amss.ac.cn>, Qingpei Hu <qingpeihu@amss.ac.cn>.

*Proceedings of the 43$^{rd}$ International Conference on Machine Learning*, Seoul, South Korea. PMLR 306, 2026. Copyright 2026 by the author(s).

et al., 2012); in engineering, the design of chemical processes or new materials aims to maximize performance metrics while satisfying safety thresholds or physical stability constraints (Sharpe et al., 2018); and in robotics, controller parameters must be tuned to maximize rewards without violating safety limits during the learning process (Berkenkamp et al., 2017). Consequently, Constrained Bayesian Optimization (CBO) has attracted significant attention.

Despite its success in low-dimensional settings, BO faces a severe challenge known as the "curse of dimensionality" (CoD). Theoretical results show that the regret bound of standard BO algorithms typically depends exponentially on the dimension $d$ of the search space (Srinivas et al., 2009). To mitigate this, Local Bayesian Optimization (LBO) has emerged as a practical compromise, aiming to find high-quality local optima rather than guaranteeing global convergence. Unlike subspace embedding (Eriksson & Jankowiak, 2021) or additive kernel (Kandasamy et al., 2015) methods which rely on structural assumptions, local methods like TuRBO (based on local trust regions, (Eriksson et al., 2019)) and GIBO (gradient-based local search, (Müller et al., 2021)), regarded as less restrictive alternatives, have demonstrated superior sample efficiency in high-dimensional unconstrained tasks.

However, scaling CBO to high dimensions remains an open challenge. Similar to the unconstrained case, the regret bounds of standard CBO methods deteriorate rapidly as dimensionality increases. For instance, EPBO (Lu & Paulson, 2022) shows that for Radial Basis Function (RBF) kernels, the regret bound scales as $\tilde{\mathcal{O}}((\log K)^{(d+1)/2} K^{-1/2})$, rendering standard global CBO inefficient for $d > 20$. While recent advances have extended local trust region methods to the constrained setting (e.g., SCBO (Eriksson & Poloczek, 2021) and HDsafeBO (Wei et al., 2025)), these approaches typically rely on rigid geometric regions to manage exploration. A critical limitation of this paradigm arises when the primary descent direction is obstructed by complex or tight constraints. In such scenarios, failing to find sufficient improvement within the current region often triggers the premature shrinkage of the trust region radius (a phenomenon we explicitly demonstrate on the truss design task in Section 6.3). In Appendix B, we provide a more detailed discussion of representative failure modes of trust-region-based CBO methods that motivate LCBO, together with em-

pirical evidence from the truss design task. Consequently, there is a notable scarcity of methods that can navigate these constrained landscapes effectively. More critically, existing high-dimensional CBO algorithms often lack rigorous theoretical guarantees regarding the dependence of convergence rates on dimensionality under mild conditions.

To address these limitations, we propose **Local Constrained Bayesian Optimization (LCBO)**, a novel framework that efficiently solves high-dimensional constrained problems. Our contributions are summarized as follows:

- We extend the local search paradigm of GIBO to the constrained setting. Unlike trust-region methods that rely on rigid geometric regions and are prone to premature shrinkage, LCBO exploits the differentiable landscape of constraint-penalized surrogates to alternate between a local "exploration" step and a local "exploitation" step to update the candidate solution.

- We provide a rigorous theoretical analysis of LCBO. Under mild assumptions, we prove that the KKT residual of the iterative sequence $\{x_k\}$ exhibits a **polynomial dependence** on the dimension $d$, effectively mitigating the curse of dimensionality found in global CBO bounds.

- We evaluate LCBO on a diverse set of high-dimensional benchmarks, including synthetic functions, engineering design problems, and reinforcement learning tasks. Our results demonstrate that LCBO achieves state-of-the-art performance, consistently outperforming existing CBO baselines.

Formally, the optimization problem is defined as:

$$\min_{\boldsymbol{x}\in\mathcal{X}} f(\boldsymbol{x}) \quad \text{s.t.} \quad c(\boldsymbol{x}) = 0, \tag{1}$$

where $f : \mathcal{X} \to \mathbb{R}$ denotes the objective function, $c : \mathcal{X} \to \mathbb{R}^m$ represents the vector-valued constraint function, and $\mathcal{X} \subset \mathbb{R}^d$ is the compact domain. We assume that the exact values of the objective and constraints are inaccessible. Instead, we interact with a noisy zero-order oracle. At time step $k$, for a query point $\boldsymbol{x}_k \in \mathcal{X}$, we observe:

$$y_{0,k} = f(\boldsymbol{x}_k) + \xi_{0,k}, \quad y_{i,k} = c_i(\boldsymbol{x}_k) + \xi_{i,k}, \tag{2}$$

where $\xi_{\cdot,k}$ are independent zero-mean Gaussian noise variables. Although our current formulation focuses on equality constraints, we note that this framework naturally extends to problems involving inequality constraints. Specifically, inequalities can be reformulated as equality constraints through the introduction of slack variables. This is a standard technique in classical optimization (Nocedal, 2006) that has been successfully applied within the context of BO (Picheny et al., 2016).

*Remark* 1.1. The CBO problem under consideration permits function evaluations in the infeasible region and considers these points to be informative and useful for updating the surrogate model, which is in contrast to safe BO (Sui et al., 2018).

## 2. Related Work

**Constrained Bayesian Optimization** Generally, strategies for extending unconstrained BO to constrained optimization problems fall into two primary categories. The first strategy involves developing an acquisition function (AF) that explicitly incorporates constraint information. Most methods in this category are modifications of vanilla AFs. A representative example is Constrained Expected Improvement (CEI) (Gardner et al., 2014), which weights the standard EI by the Probability of Feasibility (PoF). Recently, (Wang et al., 2025) proved that CEI enjoys a sub-linear convergence rate. Other well-established AFs—including Probability of Improvement (PI), Predictive Entropy Search (PES), Max-value Entropy Search (MES), and Knowledge Gradient (KG)—have also been extended to handle constrained scenarios (Carpio et al., 2018; Hernández-Lobato et al., 2015; Perrone et al., 2019; Takeno et al., 2022; Lam & Willcox, 2017; Zhang et al., 2021; Ungredda & Branke, 2024). Additionally, some approaches utilize the Upper Confidence Bound (UCB) and Lower Confidence Bound (LCB) to construct confidence intervals for the constraints. By optimizing the AF within this high-confidence domain, these methods enable explicit control over the probability of constraint violations during candidate selection (Xu et al., 2023a; Nguyen et al., 2024; Zhang & Chen, 2025).

The second strategy transforms the constrained problem into an unconstrained one, leveraging techniques from classical optimization such as penalty functions (Lu & Paulson, 2022; Pourmohamad & Lee, 2022) or (Augmented) Lagrangian methods (Picheny et al., 2016; Ariafar et al., 2019; Zhou & Ji, 2022; Xu et al., 2023b; Guo et al., 2023). In this framework, standard AFs are optimized on the resulting unconstrained surrogate problem.

**Local Bayesian Optimization** Restricting the overexploration of standard BO in high-dimensional spaces via local strategies is an effective path to mitigate the CoD. One prominent approach involves maintaining local trust regions, as seen in methods like TuRBO (Eriksson et al., 2019) and TREGO (Diouane et al., 2023). To address the CoD specifically in constrained settings, several extensions of TuRBO have been proposed for CBO (Eriksson & Poloczek, 2021; Wei et al., 2025).

Another paradigm is based on local "exploration" and "exploitation". For instance, (Müller et al., 2021) proposed a

gradient-based local search method (GIBO). Notably, (Wu et al., 2023) theoretically proved that the convergence bound of this approach depends polynomially on the dimensionality. Furthermore, (Nguyen et al., 2022) replaced the approximate gradient with the direction of maximum descent probability, while (Fan et al., 2024) demonstrated that the UCB serves as a local upper bound of the objective function and performs local search by minimizing UCB. Recent work has also explored the use of second-order information for accelerating LBO methods (Tang et al., 2025; Brunzema & Trimpe, 2026).

However, extending these local "exploration" and "exploitation" strategies to the CBO landscape remains an open problem. To address this gap, this work proposes a local search strategy specifically tailored for CBO scenarios. We further prove that, under mild conditions, the convergence bound of the proposed algorithm exhibits a polynomial dependence on the dimensionality.

# 3. Preliminaries

## 3.1. Gaussian Process Fundamentals

Gaussian Processes (GP) serve as the predominant surrogate models in BO due to their flexibility and analytical tractability. A GP is fully specified by its mean function $m(\boldsymbol{x})$ and a positive definite kernel function $k(\boldsymbol{x}, \boldsymbol{x}')$. Without loss of generality, we assume a zero-mean prior.

Given a dataset $\mathcal{D} = \{\boldsymbol{X}, \mathbf{y}\}$ comprising $n$ input locations $\boldsymbol{X} = [\boldsymbol{x}_1, \ldots, \boldsymbol{x}_n]^\top$ and corresponding noisy observations $\mathbf{y} = [y_1, \ldots, y_n]^\top$, the posterior distribution over $f$ remains a Gaussian Process. Assuming additive Gaussian noise $\epsilon \sim \mathcal{N}(0, \sigma^2)$, the posterior mean $\mu_{\mathcal{D}}(\boldsymbol{x})$ and covariance $k_{\mathcal{D}}(\boldsymbol{x}, \boldsymbol{x}')$ are derived analytically as:

$$\mu_{\mathcal{D}}(\boldsymbol{x}) = k(\boldsymbol{x}, \boldsymbol{X})\mathbf{K}^{-1}\mathbf{y}, \tag{3}$$

$$k_{\mathcal{D}}(\boldsymbol{x}, \boldsymbol{x}') = k(\boldsymbol{x}, \boldsymbol{x}') - k(\boldsymbol{x}, \boldsymbol{X})\mathbf{K}^{-1}k(\boldsymbol{X}, \boldsymbol{x}'), \tag{4}$$

where $\mathbf{K} = k(\boldsymbol{X}, \boldsymbol{X}) + \sigma^2\mathbf{I}$, $k(\boldsymbol{x}, \boldsymbol{X}) = [k(\boldsymbol{x}, \boldsymbol{x}_1), \ldots, k(\boldsymbol{x}, \boldsymbol{x}_n)]$, and $k(\boldsymbol{X}, \boldsymbol{x}) = k(\boldsymbol{x}, \boldsymbol{X})^\top$.

Utilizing differentiable kernels ensures that the function $f(\boldsymbol{x})$ and its gradient $\nabla f(\boldsymbol{x})$ follow a joint Gaussian distribution (Williams & Rasmussen, 2006). The covariance between the function and its gradient is determined by differentiating the kernel with respect to the corresponding arguments. Conditioned on the dataset $\mathcal{D}$, the posterior distribution of the gradient remains Gaussian and is given by:

$$\nabla f(\boldsymbol{x}) \mid \mathcal{D} \sim \mathcal{N}\left(\nabla\mu_{\mathcal{D}}(\boldsymbol{x}), \nabla k_{\mathcal{D}}(\boldsymbol{x}, \boldsymbol{x})\nabla^\top\right), \tag{5}$$

Here, the differential operator $\nabla$ preceding $\mu_{\mathcal{D}}(\boldsymbol{x}_k)$ and $k(\cdot, \cdot)$ denotes differentiation with respect to its first argument, whereas the operator following $k(\cdot, \cdot)$ indicates differentiation with respect to its second argument.

## 3.2. Modeling for Constrained Bayesian Optimization

In CBO we typically model the objective function $f(\boldsymbol{x})$ and $m$ constraint functions $c_i(\boldsymbol{x})$, $i = 1, \ldots, m$, with $m + 1$ independent GPs. The GP posterior then provides, for each function, both a point estimate (posterior mean) and a quantification of uncertainty (posterior variance) for the function values and their gradients.

To distinguish between the objective and constraints, we employ superscripts for the posterior statistics. The posterior mean of the objective function and its gradient are denoted by $\mu_{\mathcal{D}}^{(f)}(\boldsymbol{x})$ and $\nabla\mu_{\mathcal{D}}^{(f)}(\boldsymbol{x})$, respectively. Similarly, for the constraints, we define the vector of posterior means as $\mu_{\mathcal{D}}^{(c)}(\boldsymbol{x}) = (\mu_{\mathcal{D}}^{(c_1)}(\boldsymbol{x}), \ldots, \mu_{\mathcal{D}}^{(c_m)}(\boldsymbol{x}))^\top$ and the Jacobian matrix of the constraints as $\nabla\mu_{\mathcal{D}}^{(c)}(\boldsymbol{x}) = (\nabla\mu_{\mathcal{D}}^{(c_1)}(\boldsymbol{x}), \ldots, \nabla\mu_{\mathcal{D}}^{(c_m)}(\boldsymbol{x}))^\top$.

## 3.3. Notations and Definitions

The following notation is adopted throughout this paper. The standard inner product and the Euclidean norm are denoted by $\langle \cdot, \cdot \rangle$ and $\|\cdot\|$, respectively. For matrices, $\|\cdot\|$ specifically denotes the spectral norm. The distance from a vector $\boldsymbol{x}$ to a convex set $\Omega$ is defined as $\mathrm{d}(\boldsymbol{x}, \Omega) = \inf_{\boldsymbol{y} \in \Omega} \|\boldsymbol{y} - \boldsymbol{x}\|$. Similarly, the Euclidean projection of $\boldsymbol{x}$ onto the set $\mathcal{X}$ is denoted by $P_{\mathcal{X}}(x) = \arg\min_{\boldsymbol{y} \in \mathcal{X}} \frac{1}{2}\|\boldsymbol{y} - \boldsymbol{x}\|^2$. We denote the normal cone of a closed convex set $\mathcal{X}$ at a point $\boldsymbol{x}$ by $N_{\mathcal{X}}(\boldsymbol{x})$, which is defined as $N_{\mathcal{X}}(\boldsymbol{x}) = \{\boldsymbol{v} \in \mathbb{R}^d \mid \langle \boldsymbol{v}, \boldsymbol{y} - \boldsymbol{x} \rangle \leq 0, \forall \boldsymbol{y} \in \mathcal{X}\}$. $\mathbf{0}_{b \times d}$ denotes the zero matrix in $\mathbb{R}^{b \times d}$. Finally, we use $\mathcal{O}(\cdot)$ to denote the asymptotic upper bound, and $\tilde{\mathcal{O}}(\cdot)$ to denote the asymptotic upper bound up to logarithmic factors.

To facilitate the subsequent theoretical analysis, we provide the following definitions regarding smoothness, optimality conditions, and error metrics.

**Definition 3.1** (L-smoothness (Nesterov et al., 2018)). A differentiable function $g : \mathcal{X} \to \mathbb{R}^m$ is said to be $L$-smooth if for all $\boldsymbol{x}, \boldsymbol{y} \in \mathcal{X}$,

$$\|\nabla g(\boldsymbol{x}) - \nabla g(\boldsymbol{y})\| \leq L\|\boldsymbol{x} - \boldsymbol{y}\|, \tag{6}$$

where $\nabla g$ denotes the transpose of the Jacobian of $g$.

**Definition 3.2** (KKT Residuals (Nocedal, 2006)). To assess the quality of the approximate solutions produced by our algorithm, we quantify the violation of the Karush-Kuhn-Tucker (KKT) conditions by means of two residuals. For a candidate primal–dual pair $(\boldsymbol{x}, \lambda)$, we define the *stationarity residual*

$$r_s(\boldsymbol{x}, \lambda) := \mathrm{d}^2\big(\nabla f(\boldsymbol{x}) + \nabla c(\boldsymbol{x})^\top\lambda, -N_{\mathcal{X}}(\boldsymbol{x})\big), \tag{7}$$

that is, the distance from the vector $\nabla f(\boldsymbol{x}) + \nabla c(\boldsymbol{x})^\top\lambda$ to the negative normal cone $-N_{\mathcal{X}}(\boldsymbol{x})$. In addition, we measure

the violation of the equality constraints via the *constraint residual* or *feasibility residual*

$$r_f(\boldsymbol{x}) := \|c(\boldsymbol{x})\|^2. \tag{8}$$

Both residuals vanish simultaneously if and only if $(\boldsymbol{x}, \lambda)$ satisfies the KKT conditions, i.e., $\boldsymbol{x}$ is a first-order stationary point of the constrained problem with associated multiplier $\lambda$. In our theoretical analysis, we therefore use $r_s(\boldsymbol{x}, \lambda)$ and $r_f(\boldsymbol{x})$ to evaluate the accuracy of the solutions computed by the algorithm.

**Definition 3.3** (Error Function)**.** We define the error function $E_{d,k,\sigma}(b)$ and $e_{k,\sigma}(b)$ to bound the uncertainty in the gradient and function estimation, respectively. Formally, they are defined as:

$$E_{d,k,\sigma}(b) = \inf_{\boldsymbol{Z} \in \mathbb{R}^{b \times d}} \mathrm{tr}\big(\nabla k(\boldsymbol{0}, \boldsymbol{0})\nabla^\top - \tag{9}$$
$$\nabla k(\boldsymbol{0}, \boldsymbol{Z})(k(\boldsymbol{Z}, \boldsymbol{Z}) + \sigma^2 \boldsymbol{I})^{-1}k(\boldsymbol{Z}, \boldsymbol{0})\nabla^\top\big),$$
$$e_{k,\sigma}(b) = k(\boldsymbol{0}, \boldsymbol{0}) - k(\boldsymbol{0}, \boldsymbol{0}_{b \times d}) \tag{10}$$
$$\big(k(\boldsymbol{0}_{b \times d}, \boldsymbol{0}_{b \times d}) + \sigma^2 \boldsymbol{I}\big)^{-1}k(\boldsymbol{0}_{b \times d}, \boldsymbol{0}).$$

## 4. Proposed Algorithm

The LCBO method we propose is outlined in Algorithm 1. This approach generalizes GIBO (Müller et al., 2021) to handle constrained optimization problems defined on a compact set $\mathcal{X}$. To address the constraints, we employ a quadratic penalty method. For a penalty factor $\rho > 0$, we define the penalized objective function $Q_\rho(\boldsymbol{x})$ as:

$$Q_\rho(\boldsymbol{x}) = f(\boldsymbol{x}) + \frac{\rho}{2}\|c(\boldsymbol{x})\|^2, \tag{11}$$

where $f(\boldsymbol{x})$ is the objective function and $c(\boldsymbol{x})$ represents the constraint vector. The algorithm operates iteratively, constructing GP surrogates for both the objective and the constraints using observed data $\mathcal{D}_k$.

In the $k$-th iteration, our algorithm comprises two distinct phases: "local exploration" and "local exploitation".

In the **local exploration** phase, we actively sample data points to mitigate the uncertainty at the current design variable $\boldsymbol{x}_k$. Specifically, we select a batch of exploration candidate points $\boldsymbol{Z}_2$ (Line 6 of Algorithm 1) by minimizing the following acquisition function:

$$\alpha(\boldsymbol{x}_k, \boldsymbol{Z}) = \max_i \alpha_i(\boldsymbol{x}_k, \boldsymbol{Z}) \quad i \in \{f, c_1, \ldots, c_m\} \tag{12}$$

$$\alpha_i(\boldsymbol{x}_k, \boldsymbol{Z}) = \mathrm{tr}\big(\nabla k_{\mathcal{D}_{k-1} \cup \boldsymbol{Z}}^{(i)}(\boldsymbol{x}_k, \boldsymbol{x}_k)\nabla^\top\big) \tag{13}$$

Here, for each modelled quantity (the objective $f$ and the constraints $c_1, \ldots, c_m$), $\nabla k_{\mathcal{D}_{k-1} \cup \boldsymbol{Z}}^{(\cdot)}(\boldsymbol{x}_k, \boldsymbol{x}_k)\nabla^\top$ denotes the posterior covariance matrix of the gradient evaluated at $\boldsymbol{x}_k$ given the augmented dataset $\mathcal{D}_{k-1} \cup \boldsymbol{Z}$. This acquisition

function can be viewed as the trace of the worst-case posterior covariance matrix of the gradients at $\boldsymbol{x}_k$, conditioned on the current dataset augmented with the hypothetical batch $\boldsymbol{Z}$. By minimizing $\alpha(\boldsymbol{x}_k, \boldsymbol{Z})$ over $\boldsymbol{Z}$, we identify a batch of evaluation points that is expected to provide the largest reduction in local gradient uncertainty in a one-step lookahead sense, thereby yielding the most informative design for subsequent optimization steps. It should be noted that during the local exploration phase, we perform repeated estimations of the function value at the current design variable $\boldsymbol{x}_k$. This strategy serves a dual purpose: it reduces the posterior uncertainty of the constraints and ensures theoretical convergence.

In the **local exploitation** phase, we update the design variables by performing a gradient descent step on the penalty function Eq. (11), utilizing the posterior distribution of the GPs. The gradient is given by:

$$\hat{\nabla}Q_{\rho_k}(\boldsymbol{x}_k) = \nabla\mu_{\mathcal{D}_k}^{(f)}(\boldsymbol{x}_k) + \rho_k\nabla\mu_{\mathcal{D}_k}^{(c)}(\boldsymbol{x}_k)^\top\mu_{\mathcal{D}_k}^{(c)}(\boldsymbol{x}_k) \tag{14}$$

As the negative gradient represents the direction of steepest descent, this step can be interpreted as a localized counterpart to EI, which typically selects the point offering the maximum expected reduction in function value on a global scale.

*Remark* 4.1. As discussed in Section 3.1, the posterior quantities used in the acquisition function are not based on the exact posterior mean of the composite term $\nabla c(\boldsymbol{x})^\top c(\boldsymbol{x})$. In fact, $\nabla c(\boldsymbol{x})$ and $c(\boldsymbol{x})$ are statistically dependent under the joint GP model. We deliberately work with a simplified surrogate for two reasons. First, it leads to simpler and more efficient computations. Second, the exact posterior distribution of $\nabla c(\boldsymbol{x})^\top c(\boldsymbol{x})$ is no longer Gaussian, which makes it technically challenging to derive tight high-probability bounds on

$$\big\|\nabla c(\boldsymbol{x})^\top c(\boldsymbol{x}) - \mathbb{E}[\nabla c(\boldsymbol{x})^\top c(\boldsymbol{x}) \mid \mathcal{D}]\big\|. \tag{15}$$

By relying on a tractable Gaussian approximation instead, we obtain gradient-uncertainty measures that are easier to analyze and implement, while still capturing the main dependence structure relevant for our optimization procedure.

A distinct feature of our approach is its **single-loop** structure. Classical penalty methods typically require solving an inner unconstrained subproblem to convergence for a fixed $\rho$ before updating the penalty factor. However, in the black-box setting with noisy observations, determining the termination condition for such an inner loop is practically infeasible due to the lack of exact gradient norms. Consequently, our algorithm performs only a single gradient step per iteration. Despite this simplification, appropriate scheduling of the penalty factor $\rho_k$ and the step size $\eta_k$ ensures that the sequence of iterates converges to an approximate KKT point.

---

**Algorithm 1** Local Constrained Bayesian Optimization

---

1: **Input:** Convex and compact set $\mathcal{X}$; Black-box functions $f(\boldsymbol{x})$ and $c(\boldsymbol{x})$.
2: **Initialization:** Initial point $\boldsymbol{x}_1 \in \mathcal{X}$ and dataset $\mathcal{D}_0$; sequences for step size $\{\eta_k\}$ and penalty factor $\{\rho_k\}$.
3: **for** $k = 1, 2, \ldots, K$ **do**
4:    **Local Exploration:**
5:    Generate $\boldsymbol{Z}_1 = [\boldsymbol{x}_k, \ldots, \boldsymbol{x}_k]^\top \in \mathbb{R}^{b_k^{(1)} \times d}$ (Repeated evaluations at current iterate).
6:    Select exploration batch:

$$\boldsymbol{Z}_2 = \arg\min_{\boldsymbol{Z}} \alpha(\boldsymbol{x}_k, \boldsymbol{Z}) \quad \text{where } \boldsymbol{Z} \in \mathbb{R}^{b_k^{(2)} \times d}$$

7:    Form the combined batch $\boldsymbol{Z} = [\boldsymbol{Z}_1^\top, \boldsymbol{Z}_2^\top]^\top$.
8:    Evaluate noisy functions at $\boldsymbol{Z}$ to obtain observations $\boldsymbol{y}_f, \boldsymbol{y}_c$ and update dataset: $\mathcal{D}_k = \mathcal{D}_{k-1} \cup \{(\boldsymbol{Z}, \boldsymbol{y}_f, \boldsymbol{y}_c)\}$.
9:    **Local Exploitation:**
10:   Compute approximate gradient using GP posterior means:

$$\hat{\nabla} Q_{\rho_k}(\boldsymbol{x}_k) = \nabla \mu_{\mathcal{D}_k}^{(f)}(\boldsymbol{x}_k) + \rho_k \nabla \mu_{\mathcal{D}_k}^{(c)}(\boldsymbol{x}_k)^\top \mu_{\mathcal{D}_k}^{(c)}(\boldsymbol{x}_k)$$

11:   Update decision variable (with projection onto $\mathcal{X}$ if necessary):

$$\boldsymbol{x}_{k+1} = P_{\mathcal{X}}(\boldsymbol{x}_k - \eta_k \hat{\nabla} Q_{\rho_k}(\boldsymbol{x}_k))$$

12: **end for**

---

# 5. Theoretical Analysis

## 5.1. Assumptions

We invoke the following assumptions to establish our theoretical results:

**Assumption 5.1** (Gaussian Process Prior)**.** The objective function $f$ and constraint functions $c_i$ are sampled from zero mean GPs. Furthermore, the kernel functions are stationary, four-times continuously differentiable, positive definite and bounded.

Common kernels including RBF kernels and Matérn kernels with $\nu > 2$ satisfy the aforementioned assumptions.

**Assumption 5.2** (Domain)**.** The search space $\mathcal{X}$ is compact and convex.

**Assumption 5.3** (Regularity Condition)**.** We assume a regularity condition on the constraints that generalizes standard constraint qualifications. Specifically, there exists a constant $\gamma > 0$ such that for any sequence $\{\boldsymbol{x}_k\}$, the following inequality holds:

$$\mathrm{d}(\nabla c(\boldsymbol{x}_k)^\top c(\boldsymbol{x}_k), -N_{\mathcal{X}}(\boldsymbol{x}_k)) \geq \gamma \|c(\boldsymbol{x}_k)\|. \tag{16}$$

*Remark* 5.4. Assumption 5.3 introduces a regularity con-

dition essential for constrained optimization. Intuitively, it guarantees a steep enough descent direction towards the feasible set, preventing the optimization from stalling in infeasible regions with flat geometry. This type of condition is standard in the analysis of numerical optimization algorithms involving general constraints (Sahin et al., 2019; Li et al., 2021; Lin et al., 2022; Li et al., 2024; Alacaoglu & Wright, 2024; Lu et al., 2026). Notably, in the whole place, this condition is implied by the linear independence constraint qualification (LICQ). (Lin et al., 2022) further discusses the connection between this assumption and the Kurdyka–Łojasiewicz condition.

## 5.2. Theoretical Analysis

In this section, we establish a high-probability convergence bound for Algorithm 1, which scales polynomially with the problem dimension $d$. All detailed proofs are deferred to Appendix A.

**Theorem 5.5.** *For any $\delta \in (0, 1)$, suppose Assumptions 5.1 to 5.3 hold and the parameters in Algorithm 1 are selected according to*

$$\eta_k = \frac{1}{2L_m} k^{-\frac{1}{2}}, \quad \rho_k = k^{\frac{1}{4}}. \tag{17}$$

*Then for the total iteration count $K > \tilde{K}_m$, there exists $\lambda_{k+1}$ such that, with probability at least $1 - \delta$, the sequence $\{\boldsymbol{x}_k\}$ generated by Algorithm 1 satisfies:*

$$\frac{1}{K} \sum_{k=1}^{K} r_s(\boldsymbol{x}_{k+1}, \lambda_{k+1}) \leq \frac{C_{1,m} + C_2 \mathcal{E}_K}{\sqrt{K}}, \tag{18}$$

$$\frac{1}{K} \sum_{k=1}^{K} r_f(\boldsymbol{x}_{k+1}) \leq \frac{C_3}{\sqrt{K}} + \frac{C_{4,m} + C_5 \mathcal{E}_K}{K}. \tag{19}$$

*Here, $\mathcal{E}_K$ is the cumulative estimation error defined as:*

$$\mathcal{E}_K = \sum_{k=1}^{K} \beta_k \big( c_{1,m} e_{k,\sigma}(b_k^{(1)})$$
$$+ (c_{2,m} \tilde{B}_{k,m}^2 + c_{3,m} \eta_k) E_{d,k,\sigma}(b_k^{(2)}) \big), \tag{20}$$

*The orders of $\beta_k$ and $\tilde{B}_{k,m}$ are $\mathcal{O}(\log(mk))$ and $\mathcal{O}(\sqrt{m \log k})$, respectively. The terms $L_m, \tilde{K}_m, C_{1,m}, C_2, C_3, C_{4,m}, C_5$ and $c_{1,m}, c_{2,m}, c_{3,m}$ depend polynomially on $\gamma, m$ and the function boundaries. Detailed expressions are provided in Appendix A.3.*

Theorem 5.5 establishes that, with the exception of the cumulative error term $\mathcal{E}_K$, the remaining components of the KKT residual converges at a sublinear rate. Furthermore, there exists an index $1 \leq \hat{k} \leq K$ such that both $r_s(\boldsymbol{x}_{\hat{k}+1}, \lambda_{\hat{k}+1})$ and $r_f(\boldsymbol{x}_{\hat{k}+1})$ are of order $\mathcal{O}((1 + \mathcal{E}_K) K^{-1/2})$.

Note that $\mathcal{E}_K$ is constructed from the error functions described in Eq. (9). By further investigating the properties of these error functions, we derive the following corollary, which characterizes the polynomial dependence of the KKT residual on the dimension $d$.

**Corollary 5.6.** *Let $k(\cdot, \cdot)$ be the RBF kernel or $\nu = 2.5$ Matérn kernel. Under the same conditions as in Theorem 5.5, if we set $b_k^{(1)} = k$ and $b_k^{(2)} = dk^2$, then*

$$\frac{1}{K} \sum_{k=1}^{K} r_s(\boldsymbol{x}_{k+1}, \lambda_{k+1}) = \tilde{\mathcal{O}}(m^{\frac{7}{3}} K^{-\frac{1}{2}} + \sigma m^2 d K^{-\frac{1}{2}})$$
$$= \tilde{\mathcal{O}}(m^{\frac{5}{2}} d^{\frac{1}{6}} n^{-\frac{1}{6}} + \sigma m^{\frac{13}{6}} d^{\frac{7}{6}} n^{-\frac{1}{6}}), \quad (21)$$

$$\frac{1}{K} \sum_{k=1}^{K} r_f(\boldsymbol{x}_{k+1}) = \tilde{\mathcal{O}}(K^{-\frac{1}{2}} + m^{\frac{7}{3}} K^{-1} + \sigma m^2 d K^{-1})$$
$$= \tilde{\mathcal{O}}(m^{\frac{1}{6}} d^{\frac{1}{6}} n^{-\frac{1}{6}} + m^{\frac{8}{3}} d^{\frac{1}{3}} n^{-\frac{1}{3}} + \sigma m^{\frac{7}{3}} d^{\frac{4}{3}} n^{-\frac{1}{3}}) \quad (22)$$

*hold with probability at least $1 - \delta$. Here, $K$ denotes the iteration count and $n$ denotes the number of zero-order oracle calls.*

*Remark* 5.7. In the unconstrained optimization setting, the corresponding result is established as

$$\frac{1}{K} \sum_{k=1}^{K} r_s(\boldsymbol{x}_{k+1}) = \tilde{\mathcal{O}}(\sigma d^{\frac{4}{3}} n^{-\frac{1}{3}} + \sigma^{\frac{1}{2}} d^{\frac{2}{3}} n^{-\frac{1}{6}}) \quad (23)$$

(see Theorem 3 in (Wu et al., 2023)). Although our derived convergence bound exhibits a slightly increased dependence on the dimension $d$, this is a justifiable trade-off for the capability to handle constraints.

## 6. Experiments

In this section, we evaluate the performance of our proposed algorithm, LCBO, against state-of-the-art CBO methods. We conduct experiments on three categories of problems: (1) high-dimensional synthetic functions, (2) realistic engineering design benchmarks, and (3) a continuous control reinforcement learning task.

### 6.1. Baselines and Experimental Setup

**Baselines** We compare LCBO with four representative algorithms to demonstrate its efficacy across different aspects of constrained optimization:

- **CEI** (Gardner et al., 2014): The canonical Constrained Expected Improvement, representing standard global CBO methods.

- **EPBO** (Lu & Paulson, 2022): A method utilizing exact penalty functions, serving as a direct comparison to our penalty-based formulation.

- **SCBO** (Eriksson & Poloczek, 2021) and **HD-safeBO** (Wei et al., 2025): Two state-of-the-art approaches designed specifically for high-dimensional constrained settings.

We exclude LineBO (Kirschner et al., 2019) from our comparison as it requires a feasible starting point and a connected feasible region—assumptions that are frequently violated in the scenarios we target.

**Initialization and Protocol** We employ a cold-start strategy for all experiments. For a problem of dimension $d$, we randomly sample $d$ initial points from the search space $\mathcal{X}$ and evaluate both the objective and constraint functions. To ensure a fair comparison, all algorithms share the same initial training set $\mathcal{D}_{\text{init}}$ for GP initialization. For algorithms requiring a specific starting candidate (e.g., SCBO), we select the "best" point from $\mathcal{D}_{\text{init}}$, defined as the feasible point with the minimum objective value, or the point with the minimum constraint violation if no feasible solution exists. All experiments are repeated 10 times with different random seeds. Observations are corrupted by Gaussian noise $\epsilon \sim \mathcal{N}(0, 0.1^2)$. The optimization budget is set to 1,000 evaluations (For simplicity, we define a single evaluation as the computation of both the objective function and all constraints at a given point. Therefore, the reported evaluation count effectively corresponds to the total number of zeroth-order oracle calls divided by $(m + 1)$). Since the algorithms may not find a feasible solution immediately, and "infeasible" values cannot be averaged, we report the median and interquartile range of the best feasible objective found so far. Infeasibility is treated as infinity.

**Implementation Details** To enhance both algorithmic performance and computational efficiency, we adopt several implementation strategies following the GIBO framework, specifically: (1) maintaining a local GP model trained exclusively on the $N_m$ most recent data points; (2) normalizing $\hat{\nabla} Q_{\rho_k}(\boldsymbol{x}_k)$ using the Euclidean norm to stabilize step size; (3) restricting the search space for optimizing the acquisition function Eq. (12) to the dynamic local region $[\boldsymbol{x}_k - \delta, \boldsymbol{x}_k + \delta]$; and (4) employing a fixed mini-batch in each iteration.

Also, to enable efficient gradient-based optimization for Eq. (12), which involves maximization over multiple terms, we employ the LogSumExp function as a smooth approximation to the hard maximum operator. Detailed hyperparameter settings are provided in Appendix E. Inequality constraints are handled via the standard one-sided penalty, preserving the algorithmic mechanism (Nocedal, 2006).

## 6.2. Synthetic Benchmarks

We extend the within-model comparison framework established in GIBO (Müller et al., 2021). In this context, "within-model" implies that both the objective and constraint functions adhere to Assumption 5.1. For a comprehensive description of the synthetic objectives and specific experimental settings, we refer the reader to Section 4.1 of the GIBO paper. Our implementation diverges slightly from GIBO in that we utilize Random Fourier Features (RFF) to approximate function samples from the GP posterior (Rahimi & Recht, 2007). Furthermore, we modify the constraints to $c_i(\boldsymbol{x}) + 0.5 \le 0$. This positive offset tightens the feasible region, thereby increasing the optimization difficulty. We generate problems with dimensions $d \in \{25, 50, 100\}$ within the domain $[0, 1]^d$. Each problem includes two constraints.

Figure 1 illustrates the progressive best feasible objective values. The performance across 25D, 50D, and 100D synthetic tasks demonstrates that LCBO consistently outperforms all state-of-the-art baselines, with its advantage becoming increasingly pronounced as dimensionality scales up. Specifically, LCBO exhibits exceptional sample efficiency, achieving significantly lower objective values in the early stages of optimization compared to what other methods reach even after 1000 iterations. While the performance gap is clear in 25D, it expands dramatically in the 100D scenario, where LCBO maintains a steep descent while competitors quickly plateau, highlighting its superior scalability and robustness in navigating high-dimensional constrained search spaces.

## 6.3. Real-world Benchmarks

We evaluate the performance of our proposed method on three challenging benchmark problems: the 25-Bar Truss Design ($d = 25$), the Stepped Cantilever Beam Design ($d = 50$), and the MuJoCo HalfCheetah environment ($d = 102$). These tasks were selected to represent a diverse range of black-box constrained optimization scenarios, spanning from medium to high dimensionality. Specifically, the 25-Bar Truss and Stepped Cantilever Beam problems are canonical benchmarks in the field of structural optimization, widely used to test algorithms under complex physical constraints. Furthermore, the HalfCheetah task serves as a standard high-dimensional control benchmark from the reinforcement learning domain, demonstrating the scalability of our approach. The objective minimizes negative average forward velocity over $H = 1,000$ steps, subject to a cumulative energy constraint requiring $\sum \|a_t\|^2 \le 1,500$. Detailed descriptions of the experimental setups and problem configurations are provided in Appendix D.

Results on the **25-bar truss design task** ($d = 25$) highlight the critical advantage of our approach in boundary-constrained settings. Since the optimum lies on the constraint boundary, trust-region methods like SCBO exhibit premature stagnation—rapidly descending but then flattening out as their regions shrink against the constraints. This empirical observation corroborates our discussion in Section 1. Conversely, LCBO demonstrates continuous descent throughout the entire optimization process. This confirms that LCBO successfully mitigates the premature shrinkage issue by navigating along the constraint boundary, ultimately outperforming all baselines. Notably, EPBO exhibits poor performance on this task, remaining stuck at a very high objective level, which indicates its difficulty in effectively exploring the feasible region for this structural design problem.

For the **stepped cantilever beam task** ($d = 50$), although LCBO is slower to locate the initial feasible region (as indicated by the delayed start of the red curve compared to CEI and SCBO), it exhibits rapid and sustained convergence once feasibility is established. Unlike CEI and HDsafeBO, which suffer from obvious premature convergence and stagnate early, LCBO avoids such local optima and continues to descend even as it approaches the 1000-call limit. It is also worth noting that EPBO fails to find any feasible points within the allotted budget for this task, highlighting the superiority of LCBO in handling the tight and complex constraints typical of high-dimensional engineering challenges.

Experimental results on the **MuJoCo HalfCheetah task** ($d = 102$) demonstrate the superior sample efficiency and stability of LCBO. Regarding minor discrepancies in initial objective values, note that while all algorithms utilized identical initial input sets (averaged over five repetitions), the inherent stochasticity of the MuJoCo physics engine results in reasonable fluctuations in initial performance. During the optimization phase, LCBO exhibits the most rapid initial rate of descent, outperforming both SCBO and CEI. Although HDsafeBO achieves median performance comparable to that of LCBO between 500 and 800 evaluations, LCBO demonstrates greater overall stability across 10 independent trials and exhibits stronger continuous optimization capability as the number of function evaluations approaches 1,000. These findings substantiate the enhanced robustness and reliability of our method in complex robotic control tasks characterized by environmental noise.

## 6.4. Ablation Studies

We conduct a systematic ablation study in Appendix C to validate the key design choices of LCBO. Appendix C.1 examines the theory-practice gap arising from the growing batch size schedule assumed in our analysis, justifying the use of fixed mini-batch in the main experiments. Appendix C.2 isolates the contribution of three auxiliary components in-

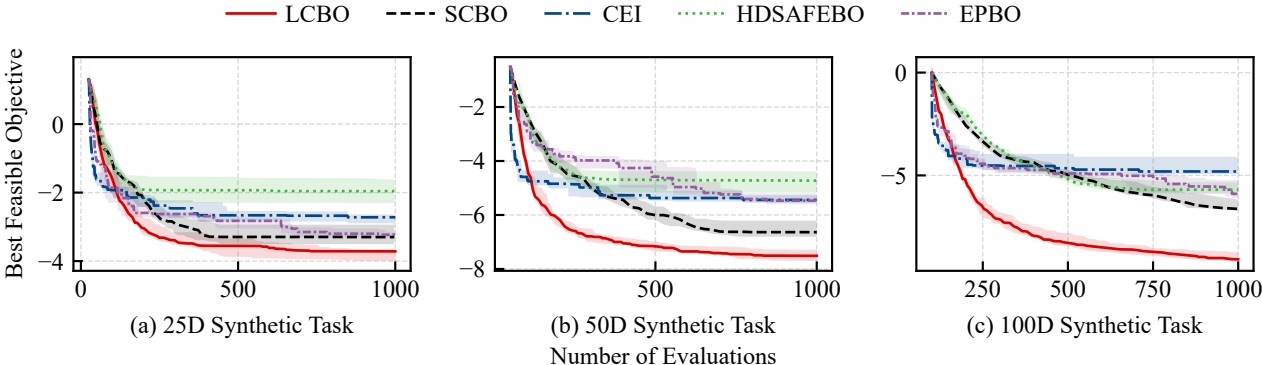

*Figure 1.* **Optimization progress on synthetic tasks.** The x-axis represents the number of function evaluations, while the y-axis shows the best feasible objective value found so far (lower is better). The solid curves represent the median performance over 10 independent runs. The shaded regions depict the interquartile range (25th–75th percentiles).

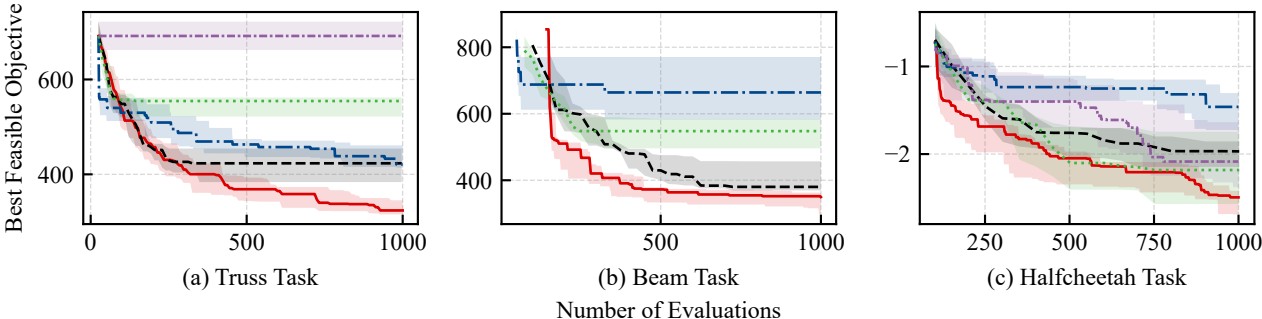

*Figure 2.* **Optimization progress on real-world tasks.** The plotting conventions and axes definitions are identical to Figure 1.

herited from GIBO including local GPs, local search boxes, and gradient normalization, which confirms that the primary gains of LCBO stem from its core penalty-based gradient descent mechanism rather than these implementation choices. Appendix C.3 investigates the sensitivity of LCBO to the penalty factor schedule $\rho_k \propto k^{1/4}$, demonstrating that LCBO performs robustly without per-task tuning.

## 7. Conclusion

In this work, we presented LCBO, a novel framework for high-dimensional CBO. By replacing the rigid trust-region mechanism with a flexible local alternating exploration-exploitation strategy, LCBO avoids premature shrinking while addressing the scalability challenges inherent in global CBO methods. Our theoretical analysis establishes that for common kernels, LCBO's KKT residual scales polynomially with dimension. This offers a more favorable scalability guarantee than the exponential dimensionality dependence typically found in the regret bounds of global CBO methods. Extensive evaluations on synthetic and real-world engineering and reinforcement learning tasks demonstrate that LCBO consistently surpasses state-of-the-art CBO base-

lines, offering superior sample efficiency, lower performance variance, and a sustained ability to refine solutions without suffering from premature convergence.

While LCBO exhibits strong local convergence, it may occasionally converge to local optima in highly multi-modal landscapes, a common trade-off in local search methods. Future work will focus on integrating global restart strategies or meta-learning techniques to enhance global exploration while maintaining the current efficiency. Overall, LCBO offers a robust and theoretically-grounded tool for solving complex, high-dimensional engineering and machine learning problems.

## Impact Statement

This paper presents work whose goal is to advance the field of Machine Learning. There are many potential societal consequences of our work, none which we feel must be specifically highlighted here.

## Acknowledgement

Jingzhe Jing, Zheyi Fan, and Qingpei Hu's work is supported by the National Key Research and Development Program of China (2021YFA1000300 and 2021YFA1000301).

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

# A. Proof of Main Theorem and Corollary

## A.1. Properties of Gaussian Process and its Derivative

In this section, we present lemmas establishing high-probability error bounds for the posterior mean estimator $\mu_{\mathcal{D}_k}(\boldsymbol{x}_k)$ and $\nabla\mu_{\mathcal{D}_k}(\boldsymbol{x}_k)$. Note that the GP posterior covariance does not depend on the observed responses but only on the design points and the kernel hyperparameters. In the coupled setting where the objective and all constraints are evaluated at the same inputs, their posterior variances are identical, differing only by their hyperparameters (assuming all GPs utilize the same kernel function). Since our primary focus is on an order-of-magnitude analysis, for the sake of notational simplicity in the following proof, we will not use superscripts to distinguish between different error functions; instead, we will uniformly denote them as $E_{d,k,\sigma}(b_k)$ and $e_{k,\sigma}(b_k)$ for the gradient and function-value error functions, respectively. In addition, let $\mathrm{tr}\left(\nabla k_{\mathcal{D}_k}(\boldsymbol{x}_k, \boldsymbol{x}_k)\nabla^\top\right)$ and $\mathrm{tr}\left(k_{\mathcal{D}_k}(\boldsymbol{x}_k, \boldsymbol{x}_k)\right)$ denote $\max_i\{\mathrm{tr}\left(\nabla k_{\mathcal{D}_k}^{(i)}(\boldsymbol{x}_k, \boldsymbol{x}_k)\nabla^\top\right)\}$ and $\max_i\{k_{\mathcal{D}_k}^{(i)}(\boldsymbol{x}_k, \boldsymbol{x}_k)\}, i \in \{f, c_1, \ldots, c_m\}$ respectively.

**Lemma A.1.** *Let $\mathcal{X} \subset \mathbb{R}^d$ be compact and convex, and let $g : \mathcal{X} \to \mathbb{R}$ be a zero-mean GP with kernel $k \in \mathcal{C}^4(\mathcal{X} \times \mathcal{X})$. Set*

$$\mathcal{T} := \mathcal{X} \times \mathbb{S}^{d-1} \times \mathbb{S}^{d-1}, \qquad Z(\boldsymbol{x}, \boldsymbol{u}, \boldsymbol{v}) := \boldsymbol{u}^\top \nabla^2 g(\boldsymbol{x})\, \boldsymbol{v},$$

*and let*

$$\sigma_*^2 := \sup_{(\boldsymbol{x},\boldsymbol{u},\boldsymbol{v})\in\mathcal{T}} \mathrm{Var}\big(Z(\boldsymbol{x}, \boldsymbol{u}, \boldsymbol{v})\big), \qquad M := \mathbb{E}\Big[\sup_{(\boldsymbol{x},\boldsymbol{u},\boldsymbol{v})\in\mathcal{T}} |Z(\boldsymbol{x}, \boldsymbol{u}, \boldsymbol{v})|\Big].$$

*Then for any $\delta \in (0,1)$, with probability at least $1 - \delta$,*

$$\sup_{\boldsymbol{x}\in\mathcal{X}} \|\nabla^2 g(\boldsymbol{x})\| \le M + \sigma_* \sqrt{2\log\frac{2}{\delta}}. \tag{24}$$

*Consequently, $\nabla g$ is $L$-Lipschitz continuous on $\mathcal{X}$, with*

$$L := M + \sigma_* \sqrt{2\log\frac{2}{\delta}}.$$

*Proof.* Since $k \in \mathcal{C}^4$, all mixed partial derivatives of order up to four exist and are continuous. This ensures that $g$ admits a modification with almost surely continuous first and second derivatives on $\mathcal{X}$, so $\nabla^2 g(x)$ is well-defined and continuous almost surely (Theorem 1.4.2. in (Adler & Taylor, 2007)). For any $x \in \mathcal{X}$,

$$\|\nabla^2 g(\boldsymbol{x})\| = \sup_{\boldsymbol{u},\boldsymbol{v}\in\mathbb{S}^{d-1}} |\boldsymbol{u}^\top \nabla^2 g(\boldsymbol{x})\, \boldsymbol{v}|,$$

hence

$$\sup_{\boldsymbol{x}\in\mathcal{X}} \|\nabla^2 g(\boldsymbol{x})\| = \sup_{(\boldsymbol{x},\boldsymbol{u},\boldsymbol{v})\in\mathcal{T}} |Z(\boldsymbol{x}, \boldsymbol{u}, \boldsymbol{v})|.$$

The process $Z$ is Gaussian, centered, and continuous on the compact index set $\mathcal{T}$. Its expected supremum $M$ is finite. Its variance is uniformly bounded by $\sigma_*^2 < +\infty$ because the fourth derivatives of $k$ are bounded on $\mathcal{X} \times \mathcal{X}$.

Applying the Borell–TIS inequality to $U := \sup_{(\boldsymbol{x},\boldsymbol{u},\boldsymbol{v})\in\mathcal{T}} |Z(\boldsymbol{x}, \boldsymbol{u}, \boldsymbol{v})|$ yields

$$\mathbb{P}(U \ge M + t) \le 2\exp\left(-\frac{t^2}{2\sigma_*^2}\right).$$

Choosing $t = \sigma_* \sqrt{2\log(2/\delta)}$ gives Eq. (24).

Finally, by the mean-value inequality for vector-valued functions,

$$\|\nabla g(\boldsymbol{x}) - \nabla g(\boldsymbol{y})\| \le L\|\boldsymbol{x} - \boldsymbol{y}\| \quad \text{for all } \boldsymbol{x}, \boldsymbol{y} \in \mathcal{X},$$

so $\nabla g$ is $L$-Lipschitz continuous. This completes the proof. $\square$

**Corollary A.2.** *Suppose Assumptions 5.1 and 5.2 hold. Then for any $\delta \in (0, 1)$, there exist a constant $L_{\nabla f} > 0$ and a base constant $\tilde{L}_{\nabla c} > 0$ (weakly dependent on $\log m$) such that, defining $L_{\nabla c, m} := \sqrt{m} \tilde{L}_{\nabla c}$, with probability at least $1 - \delta/6$, $f$ and $c$ are $L_{\nabla f}$- and $L_{\nabla c, m}$-smooth on $\mathcal{X}$, respectively. That is, for all $\boldsymbol{x}, \boldsymbol{y} \in \mathcal{X}$,*

$$\|\nabla f(\boldsymbol{x}) - \nabla f(\boldsymbol{y})\| \le L_{\nabla f} \|\boldsymbol{x} - \boldsymbol{y}\|, \tag{25}$$
$$\|\nabla c(\boldsymbol{x}) - \nabla c(\boldsymbol{y})\| \le L_{\nabla c, m} \|\boldsymbol{x} - \boldsymbol{y}\|. \tag{26}$$

*Proof.* The proof proceeds by bounding the spectral norm of the Hessians of the Gaussian processes using the Borell-TIS inequality.

For $f$, choose a failure budget $\delta_f := \delta/12$. Let $M_f$ and $\sigma_{*,f}$ be the mean-envelope and variance-envelope constants of the Hessian process of $f$ (as in Lemma A.1). We obtain

$$\sup_{\boldsymbol{x} \in \mathcal{X}} \|\nabla^2 f(\boldsymbol{x})\| \le L_{\nabla f} := M_f + \sigma_{*,f} \sqrt{2 \log \frac{2}{\delta_f}}$$

with probability at least $1 - \delta_f$.

For the vector-valued function $c$, we analyze its components $c_i$ ($i = 1, \ldots, m$). We allocate a failure budget $\delta_i := \delta/(12m)$ to each component. Let $M_c := \max_i M_i$ and $\sigma_{*,c} := \max_i \sigma_{*,i}$ be the maximum envelope constants across all components. With probability at least $1 - m \cdot \delta_i = 1 - \delta/12$, all components satisfy:

$$\sup_{\boldsymbol{x} \in \mathcal{X}} \|\nabla^2 c_i(\boldsymbol{x})\| \le \tilde{L}_{\nabla c} := M_c + \sigma_{*,c} \sqrt{2 \log \frac{24m}{\delta}}, \quad \forall i = 1, \ldots, m.$$

Thus, each $c_i$ is $\tilde{L}_{\nabla c}$-smooth. To obtain the Lipschitz constant for the Jacobian $\nabla c$, we bound its spectral norm by the Frobenius norm. Note that $\nabla c(\boldsymbol{x}) - \nabla c(\boldsymbol{y})$ is a matrix whose rows are $(\nabla c_i(\boldsymbol{x}) - \nabla c_i(\boldsymbol{y}))^\top$.

$$\|\nabla c(\boldsymbol{x}) - \nabla c(\boldsymbol{y})\| \le \|\nabla c(\boldsymbol{x}) - \nabla c(\boldsymbol{y})\|_F$$
$$= \sqrt{\sum_{i=1}^{m} \|\nabla c_i(\boldsymbol{x}) - \nabla c_i(\boldsymbol{y})\|^2}$$
$$\le \sqrt{\sum_{i=1}^{m} \tilde{L}_{\nabla c}^2 \|\boldsymbol{x} - \boldsymbol{y}\|^2}$$
$$= \sqrt{m} \tilde{L}_{\nabla c} \|\boldsymbol{x} - \boldsymbol{y}\|.$$

We define $L_{\nabla c, m} := \sqrt{m} \tilde{L}_{\nabla c}$. Combining the failure probabilities for $f$ and $c$ ($\delta/12 + \delta/12 = \delta/6$), the joint smoothness holds with probability at least $1 - \delta/6$. $\qquad\square$

**Corollary A.3.** *Under the same condition as Corollary A.2, for any $\delta \in (0, 1)$, there exist constants $B_f, B_{\nabla f} > 0$ and base constants $\tilde{B}_c, \tilde{B}_{\nabla c} > 0$ (which scale logarithmically with $m$) such that, defining $B_{c,m} := \sqrt{m} \tilde{B}_c$ and $B_{\nabla c, m} := \sqrt{m} \tilde{B}_{\nabla c}$, with probability at least $1 - \delta/6$, the following bounds hold:*

$$\sup_{\boldsymbol{x} \in \mathcal{X}} |f(\boldsymbol{x})| \le B_f, \qquad \sup_{\boldsymbol{x} \in \mathcal{X}} \|c(\boldsymbol{x})\| \le B_{c,m}. \tag{27}$$
$$\sup_{\boldsymbol{x} \in \mathcal{X}} \|\nabla f(\boldsymbol{x})\| \le B_{\nabla f}, \qquad \sup_{\boldsymbol{x} \in \mathcal{X}} \|\nabla c(\boldsymbol{x})\| \le B_{\nabla c, m} \tag{28}$$

*Proof.* The proof follows the same strategy as Lemma A.1 by applying the Borell-TIS inequality. We allocate the total failure budget $\delta/6$ across the four terms.

Bounding $f$ and $c$:

Consider the centered GP $f$ on the compact set $\mathcal{X}$. Let $M_{f,0} := \mathbb{E}[\sup_{\boldsymbol{x} \in \mathcal{X}} |f(\boldsymbol{x})|]$ and $\sigma_{f,0}^2 := \sup_{\boldsymbol{x} \in \mathcal{X}} \text{Var}(f(\boldsymbol{x}))$. By the Borell-TIS inequality, with probability at least $1 - \delta/24$,

$$\sup_{\boldsymbol{x} \in \mathcal{X}} |f(\boldsymbol{x})| \le B_f := M_{f,0} + \sigma_{f,0} \sqrt{2 \log \frac{48}{\delta}}.$$

For the vector-valued function $c(\boldsymbol{x}) = [c_1(\boldsymbol{x}), \ldots, c_m(\boldsymbol{x})]^\top$, we apply the bound component-wise. We allocate a failure probability of $\delta/(24m)$ to each component $c_i$. Let $M_{c,0} = \max_i \mathbb{E}[\sup_{\boldsymbol{x}} |c_i(\boldsymbol{x})|]$ and $\sigma_{c,0}^2 = \max_i \sup_{\boldsymbol{x}} \mathrm{Var}(c_i(\boldsymbol{x}))$. With probability at least $1 - m \cdot \frac{\delta}{24m} = 1 - \delta/24$, all components satisfy:

$$\sup_{\boldsymbol{x} \in \mathcal{X}} |c_i(\boldsymbol{x})| \le \tilde{B}_c := M_{c,0} + \sigma_{c,0}\sqrt{2\log\frac{48m}{\delta}}, \quad \forall i = 1, \ldots, m.$$

Consequently, the Euclidean norm of $c(\boldsymbol{x})$ is bounded by:

$$\sup_{\boldsymbol{x} \in \mathcal{X}} \|c(\boldsymbol{x})\| = \sup_{\boldsymbol{x} \in \mathcal{X}} \sqrt{\sum_{i=1}^m |c_i(\boldsymbol{x})|^2} \le \sqrt{\sum_{i=1}^m \tilde{B}_c^2} = \sqrt{m}\tilde{B}_c =: B_{c,m}.$$

Bounding $\nabla f$ and $\nabla c$:

To bound the gradient norm, we define the scalarized process on $\mathcal{T}_1 := \mathcal{X} \times \mathbb{S}^{d-1}$ as $Z_{\nabla f}(\boldsymbol{x}, \boldsymbol{u}) := \boldsymbol{u}^\top \nabla f(\boldsymbol{x})$. Using parameters $M_{f,1}$ and $\sigma_{f,1}^2$ for this process, Borell-TIS yields that with probability at least $1 - \delta/24$:

$$\sup_{\boldsymbol{x} \in \mathcal{X}} \|\nabla f(\boldsymbol{x})\| \le B_{\nabla f} := M_{f,1} + \sigma_{f,1}\sqrt{2\log\frac{48}{\delta}}.$$

For the Jacobian $\nabla c(\boldsymbol{x})$, we bound the spectral norm via the Frobenius norm: $\|\nabla c(\boldsymbol{x})\| \le \|\nabla c(\boldsymbol{x})\|_F = \sqrt{\sum_{i=1}^m \|\nabla c_i(\boldsymbol{x})\|^2}$. Similar to the case for $c$, we apply Borell-TIS to each gradient component $\nabla c_i$ (considered as a vector field) with failure budget $\delta/(24m)$. Let $\tilde{B}_{\nabla c}$ be the uniform upper bound for $\sup_{\boldsymbol{x}} \|\nabla c_i(\boldsymbol{x})\|$. With probability at least $1 - \delta/24$, we have $\sup_{\boldsymbol{x}} \|\nabla c_i(\boldsymbol{x})\| \le \tilde{B}_{\nabla c}$ for all $i$, which implies:

$$\sup_{\boldsymbol{x} \in \mathcal{X}} \|\nabla c(\boldsymbol{x})\| \le \sqrt{\sum_{i=1}^m \tilde{B}_{\nabla c}^2} = \sqrt{m}\tilde{B}_{\nabla c} =: B_{\nabla c,m}.$$

Summing the failure probabilities, the bounds hold simultaneously with probability at least $1 - \delta/6$. $\qquad \square$

**Lemma A.4.** *Fix* $0 < \delta < 1$ *and define* $\beta_k' = 2\log\big(\pi^2 k^2/(3\delta)\big)$. *Let* $(\boldsymbol{u}_k)_{k \ge 1}$ *be random variables such that for each* $k$, $\boldsymbol{u}_k \mid \mathcal{D}_k \sim \mathcal{N}(0, \Sigma_k)$. *Then with probability at least* $1 - \delta$, *for all* $k \ge 1$ *simultaneously,*

$$\|\boldsymbol{u}_k\|^2 \le \beta_k' tr(\Sigma_k). \tag{29}$$

*Proof.* Fix $k \ge 1$. Conditional on $\mathcal{D}_k$, we have $\boldsymbol{u}_k = \Sigma_k^{1/2} g$ with $g \sim \mathcal{N}(0, I)$. By a standard Gaussian quadratic-form tail bound (lemma 10 in (Wu et al., 2023) applied conditional on $\mathcal{D}_k$),

$$\mathbb{P}\big(\|\boldsymbol{u}_k\|^2 \ge \beta_k' \operatorname{tr}(\Sigma_k) \,\big|\, \mathcal{D}_k\big) \le 2\exp\big(-\frac{\beta_k'}{2}\big).$$

With the above choice of $\beta_k'$, the right-hand side equals $6\delta/(\pi^2 k^2)$. Taking a union bound over all $k \ge 1$ and using $\sum_{k=1}^\infty k^{-2} = \pi^2/6$ yield

$$\mathbb{P}\big(\exists k \ge 1 : \|\boldsymbol{u}_k\|^2 \ge \beta_k \operatorname{tr}(\Sigma_k)\big) \le \sum_{k=1}^\infty \frac{6\delta}{\pi^2 k^2} = \delta,$$

$\qquad \square$

**Corollary A.5.** *Suppose Assumptions 5.1 and 5.2 hold. On the event that* $\sup_{\boldsymbol{x} \in \mathcal{X}} \|c(\boldsymbol{x})\| \le B_{c,m} < +\infty$, *for any* $\delta \in (0,1)$ *and* $k \ge 1$, *with probability at least* $1 - \delta/3$,

$$\sup_{\boldsymbol{x} \in \mathcal{X}} \|\mu_{\mathcal{D}_k}^{(c)}(\boldsymbol{x})\| \le \tilde{B}_{k,m}, \tag{30}$$

*where* $\tilde{B}_{k,m} = B_{c,m} + \sqrt{2m\kappa \log\big(\pi^2 k^2/\delta\big)}$ *and* $\kappa = \max_i \sup_{\boldsymbol{x} \in \mathcal{X}} k^{(c_i)}(\boldsymbol{x}, \boldsymbol{x}) < +\infty, i = 1, 2, \cdots, m$.

*Proof.* For a fixed $k \geq 1$, let $\boldsymbol{x}_k^\star \in \arg\max_{\boldsymbol{x} \in \mathcal{X}} \|\mu_{\mathcal{D}_k}^{(c)}(\boldsymbol{x})\|$. Conditioned on the given $\mathcal{D}_k$, the point $\boldsymbol{x}_k^\star$ is deterministic. By the triangle inequality,

$$\|\mu_{\mathcal{D}_k}^{(c)}(\boldsymbol{x}_k^\star)\| \leq \|\boldsymbol{c}(\boldsymbol{x}_k^\star)\| + \|\mu_{\mathcal{D}_k}^{(c)}(\boldsymbol{x}_k^\star) - \boldsymbol{c}(\boldsymbol{x}_k^\star)\| \leq \sup_{\boldsymbol{x} \in \mathcal{X}} \|\boldsymbol{c}(\boldsymbol{x})\| + \|\mu_{\mathcal{D}_k}^{(c)}(\boldsymbol{x}_k^\star) - \boldsymbol{c}(\boldsymbol{x}_k^\star)\|. \tag{31}$$

Specifically, the difference $\mu_{\mathcal{D}_k}^{(c)}(\boldsymbol{x}_k^\star) - \boldsymbol{c}(\boldsymbol{x}_k^\star)$ follows a $\mathcal{N}(\boldsymbol{0}, \Sigma_k)$ distribution, satisfying the condition that $\mathrm{tr}(\Sigma_k) \leq m\kappa$. Applying Lemma A.4, in conjunction with Eq. (31), we establish that with probability at least $1 - \delta/3$, the following holds for any $k \geq 1$:

$$\sup_{\boldsymbol{x} \in \mathcal{X}} \|\mu_{\mathcal{D}_k}^{(c)}(\boldsymbol{x})\| \leq \sup_{\boldsymbol{x} \in \mathcal{X}} \|\boldsymbol{c}(\boldsymbol{x})\| + \|\mu_{\mathcal{D}_k}^{(c)}(\boldsymbol{x}_k^\star) - \boldsymbol{c}(\boldsymbol{x}_k^\star)\|$$

$$\leq B_{c,m} + t_k.$$

where $t_{k,m} = \sqrt{2m\kappa \log(\pi^2 k^2/\delta)}$. Setting $\tilde{B}_{k,m} = B_{c,m} + t_{k,m}$ concludes the proof. $\qquad\square$

**Corollary A.6.** *For any $0 < \delta < 1$, let $\beta_k = 2\log\big(2(m+1)\pi^2 k^2/\delta\big)$. Then, for any iteration $k \geq 1$, the following bounds hold:*

$$\big|f(\boldsymbol{x}_k) - \mu_{\mathcal{D}_k}^{(f)}(\boldsymbol{x}_k)\big|^2 \leq \beta_k\, k_{\mathcal{D}_k}(\boldsymbol{x}_k, \boldsymbol{x}_k), \qquad \text{with prob. at least } 1 - \delta/(6m+6), \tag{32}$$

$$\big\|\nabla f(\boldsymbol{x}_k) - \nabla\mu_{\mathcal{D}_k}^{(f)}(\boldsymbol{x}_k)\big\|^2 \leq \beta_k \mathrm{tr}\big(\nabla k_{\mathcal{D}_k}(\boldsymbol{x}_k, \boldsymbol{x}_k)\nabla^\top\big), \qquad \text{with prob. at least } 1 - \delta/(6m+6), \tag{33}$$

$$\big\|c(\boldsymbol{x}_k) - \mu_{\mathcal{D}_k}^{(c)}(\boldsymbol{x}_k)\big\|^2 \leq m\,\beta_k\, k_{\mathcal{D}_k}(\boldsymbol{x}_k, \boldsymbol{x}_k), \qquad \text{with prob. at least } 1 - m\delta/(6m+6), \tag{34}$$

$$\big\|\nabla c(\boldsymbol{x}_k) - \nabla\mu_{\mathcal{D}_k}^{(c)}(\boldsymbol{x}_k)\big\|^2 \leq m\,\beta_k \mathrm{tr}\big(\nabla k_{\mathcal{D}_k}(\boldsymbol{x}_k, \boldsymbol{x}_k)\nabla^\top\big), \qquad \text{with prob. at least } 1 - m\delta/(6m+6). \tag{35}$$

*Proof.* The first and second inequalities (Eq. (32) and Eq. (33)) follow by applying Lemma A.4.

For the constraints, applying the same bound to each component $c_i(\boldsymbol{x}_k) - \mu_{\mathcal{D}_k}^{(c_i)}(\boldsymbol{x}_k)$ and $\nabla c_i(\boldsymbol{x}_k) - \nabla\mu_{\mathcal{D}_k}^{(c_i)}(\boldsymbol{x}_k)$ yields, for $i \in \{1, \ldots, m\}$,

$$\big|c_i(\boldsymbol{x}_k) - \mu_{\mathcal{D}_k}^{(c_i)}(\boldsymbol{x}_k)\big|^2 \leq \beta_k\, k_{\mathcal{D}_k}(\boldsymbol{x}_k, \boldsymbol{x}_k), \quad \big\|\nabla c_i(\boldsymbol{x}_k) - \nabla\mu_{\mathcal{D}_k}^{(c_i)}(\boldsymbol{x}_k)\big\|^2 \leq \beta_k\, \mathrm{tr}\big(\nabla k_{\mathcal{D}_k}(\boldsymbol{x}_k, \boldsymbol{x}_k)\nabla^\top\big),$$

each with probability at least $1 - \delta/(6m+6)$. A union bound over $i = 1, \ldots, m$ gives probability at least $1 - m\delta/(6m+6)$ that all component-wise bounds hold simultaneously. On this event, we have

$$\big\|c(\boldsymbol{x}_k) - \mu_{\mathcal{D}_k}^{(c)}(\boldsymbol{x}_k)\big\|^2 = \sum_{i=1}^m \big|c_i(\boldsymbol{x}_k) - \mu_{\mathcal{D}_k}^{(c_i)}(\boldsymbol{x}_k)\big|^2 \leq m\,\beta_k\, k_{\mathcal{D}_k}(\boldsymbol{x}_k, \boldsymbol{x}_k),$$

which proves Eq. (34). Moreover, letting $A := \nabla c(\boldsymbol{x}_k) - \nabla\mu_{\mathcal{D}_k}^{(c)}(\boldsymbol{x}_k) \in \mathbb{R}^{m \times d}$, we first bound its Frobenius norm:

$$\|A\|_F^2 = \sum_{i=1}^m \big\|\nabla c_i(\boldsymbol{x}_k) - \nabla\mu_{\mathcal{D}_k}^{(c_i)}(\boldsymbol{x}_k)\big\|^2 \leq m\,\beta_k\, \mathrm{tr}\big(\nabla k_{\mathcal{D}_k}(\boldsymbol{x}_k, \boldsymbol{x}_k)\nabla^\top\big).$$

Finally, since $\|A\| \leq \|A\|_F$ for the spectral norm $\|\cdot\|$, we obtain

$$\big\|\nabla c(\boldsymbol{x}_k) - \nabla\mu_{\mathcal{D}_k}^{(c)}(\boldsymbol{x}_k)\big\|^2 = \|A\|^2 \leq \|A\|_F^2 \leq m\,\beta_k\, \mathrm{tr}\big(\nabla k_{\mathcal{D}_k}(\boldsymbol{x}_k, \boldsymbol{x}_k)\nabla^\top\big),$$

which proves Eq. (35). $\qquad\square$

*Remark* A.7. Let $E_1, \ldots, E_7$ denote the events where Eq. (25), Eq. (27), Eq. (30), Eq. (32), Eq. (33), Eq. (34) and Eq. (35) hold, respectively, and define $E = \cap_{i=1}^7 E_i$. According to Corollary A.5, we have $\mathbb{P}(E_3 \mid E_2) \geq 1 - \delta/3$; consequently, $\mathbb{P}(E_2 E_3) = \mathbb{P}(E_3 \mid E_2)\mathbb{P}(E_2) \geq 1 - \delta/2$. Then it follows that

$$\mathbb{P}(E) = \mathbb{P}(\cap_{i=1}^7 E_i) = 1 - \mathbb{P}(\cup_{i=1}^7 E_i^c) \geq 1 - \big(\mathbb{P}(E_1) + \mathbb{P}((E_2 E_3)^c) + \sum_{i=4}^7 \mathbb{P}(E_i^c)\big)$$

$$\geq 1 - \frac{\delta}{6} - \frac{\delta}{2} - \frac{(2m+2)\delta}{6m+6}$$

$$= 1 - \delta.$$

For notational convenience, the subsequent discussion proceeds conditioned on the occurrence of event $E$.

Furthermore, let the constant $L_m$ be defined as

$$L_m = L_{\nabla f} + B_{c,m} L_{\nabla c,m} + B_{\nabla c,m}^2$$

and $L_{\rho,m} = \rho L_m$. It is straightforward to verify that the penalty function $Q_\rho(\cdot)$ is $L_{\rho,m}$-smooth (assuming $\rho \geq 1$).

Specifically, note that from Corollary A.3, $\sup_{\boldsymbol{x}} \|\nabla c(\boldsymbol{x})\| \leq B_{\nabla c,m}$, which implies that $c$ is $B_{\nabla c,m}$-Lipschitz continuous. Then for any $\boldsymbol{x}, \boldsymbol{y} \in \mathcal{X}$:

$$\begin{aligned}
&\|\nabla Q_\rho(\boldsymbol{x}) - \nabla Q_\rho(\boldsymbol{y})\| \\
&\leq \|\nabla f(\boldsymbol{x}) - \nabla f(\boldsymbol{y})\| + \rho \|\nabla c(\boldsymbol{x})^\top c(\boldsymbol{x}) - \nabla c(\boldsymbol{y})^\top c(\boldsymbol{y})\| \\
&\leq \|\nabla f(\boldsymbol{x}) - \nabla f(\boldsymbol{y})\| + \rho \|\nabla c(\boldsymbol{x})^\top (c(\boldsymbol{x}) - c(\boldsymbol{y})) + (\nabla c(\boldsymbol{x}) - \nabla c(\boldsymbol{y}))^\top c(\boldsymbol{y})\| \\
&\leq L_{\nabla f} \|\boldsymbol{x} - \boldsymbol{y}\| + \rho \left( \|\nabla c(\boldsymbol{x})\| \|c(\boldsymbol{x}) - c(\boldsymbol{y})\| + \|\nabla c(\boldsymbol{x}) - \nabla c(\boldsymbol{y})\| \|c(\boldsymbol{y})\| \right) \\
&\leq L_{\nabla f} \|\boldsymbol{x} - \boldsymbol{y}\| + \rho \left( B_{\nabla c,m} \cdot B_{\nabla c,m} \|\boldsymbol{x} - \boldsymbol{y}\| + L_{\nabla c,m} \|\boldsymbol{x} - \boldsymbol{y}\| \cdot B_{c,m} \right) \\
&= \left( L_{\nabla f} + \rho(B_{\nabla c,m}^2 + B_{c,m} L_{\nabla c,m}) \right) \|\boldsymbol{x} - \boldsymbol{y}\| \\
&\leq \rho \left( L_{\nabla f} + B_{\nabla c,m}^2 + B_{c,m} L_{\nabla c,m} \right) \|\boldsymbol{x} - \boldsymbol{y}\| \quad \text{(since } \rho \geq 1) \\
&= L_{\rho,m} \|\boldsymbol{x} - \boldsymbol{y}\|.
\end{aligned}$$

**Lemma A.8.** *In the $k$-th iteration of Algorithm 1, we have:*

$$tr(\nabla k_{\mathcal{D}_k}(\boldsymbol{x}_k, \boldsymbol{x}_k) \nabla^\top) \leq E_{d,k,\sigma}(b_k^{(2)}). \tag{36}$$

*Proof.* During the $k$-th iteration of Algorithm 1, the candidate set can be partitioned into two components. Specifically, one component is derived by optimizing Eq. (12). Let us denote this subset of data as $\{\boldsymbol{Z}_2, \boldsymbol{y}_2\}$ and define $\mathcal{D}_k' = \mathcal{D}_{k-1} \cup \{\boldsymbol{Z}_2, \boldsymbol{y}_2\}$. Given that $\mathcal{D}_k'$ is a subset of $\mathcal{D}_k$, it follows that:

$$\text{tr}(\nabla k_{\mathcal{D}_k}(\boldsymbol{x}_k, \boldsymbol{x}_k) \nabla^\top) \leq \text{tr}(\nabla k_{\mathcal{D}_k'}(\boldsymbol{x}_k, \boldsymbol{x}_k) \nabla^\top).$$

By applying an analysis analogous to Lemma 8 in (Wu et al., 2023), we can establish that $\text{tr}(\nabla k_{\mathcal{D}_k'}(\boldsymbol{x}_k, \boldsymbol{x}_k) \nabla^\top) \leq E_{d,k,\sigma}(b_k^{(2)})$ which yields the desired conclusion. $\square$

**Lemma A.9.** *In the $k$-th iteration of Algorithm 1, we have:*

$$k_{\mathcal{D}_k}(\boldsymbol{x}_k, \boldsymbol{x}_k) \leq e_{k,\sigma}(b_k^{(1)}). \tag{37}$$

*Proof.* The proof follows a procedure identical to that of Lemma A.8, with the sole modification being the substitution of $\text{tr}(\nabla k_{\mathcal{D}_k}(\boldsymbol{x}_k, \boldsymbol{x}_k)$ and $E_{d,k,\sigma}(b_k^{(2)})$ with $k_{\mathcal{D}_k}(\boldsymbol{x}_k, \boldsymbol{x}_k)$ and $e_{k,\sigma}(b_k^{(1)})$, respectively. $\square$

**Lemma A.10.** *Let $k(\boldsymbol{x}_1, \boldsymbol{x}_2)$ be the RBF kernel or the $\nu = 2.5$ Matérn kernel. We have:*

$$E_{d,k,\sigma}(b) \lesssim \mathcal{O}(\sigma d^{\frac{3}{2}} b^{-\frac{1}{2}}). \tag{38}$$

*Proof.* See Lemmas 4 and 5 in (Wu et al., 2023). $\square$

**Lemma A.11.** *For any kernel function satisfying $k(\boldsymbol{0}, \boldsymbol{0}) = \kappa_0 < +\infty$, we have:*

$$e_{k,\sigma}(b) \lesssim \mathcal{O}(\sigma^2 b^{-1}). \tag{39}$$

*Proof.* Let $\boldsymbol{Z} = (\boldsymbol{z}_1, \ldots, \boldsymbol{z}_b) = (\boldsymbol{0}, \boldsymbol{0}, \ldots, \boldsymbol{0})$. We analyze the variance reduction term $A(\boldsymbol{Z})$:

$$\begin{aligned}
A(\boldsymbol{Z}) &= k(\boldsymbol{0}, \boldsymbol{0}) - k(\boldsymbol{0}, \boldsymbol{Z})(k(\boldsymbol{Z}, \boldsymbol{Z}) + \sigma^2 I_b)^{-1} k(\boldsymbol{Z}, \boldsymbol{0}) \\
&= \kappa_0 \left( 1 - \mathbf{1}^\top (\mathbf{1}\mathbf{1}^\top + \sigma^2 I_b)^{-1} \mathbf{1} \right).
\end{aligned}$$

Using the rank-1 update inverse, we have:

$$(\mathbf{1}\mathbf{1}^\top + \sigma^2 I_b)^{-1} = \frac{1}{\sigma^2} I_b - \frac{1}{\sigma^2(\sigma^2 + b)}\mathbf{1}\mathbf{1}^\top.$$

Substituting this back into the expression for $A(\mathbf{Z})$:

$$\begin{aligned}
A(\mathbf{Z}) &= \kappa_0 \left(1 - \mathbf{1}^\top \big(\frac{I_b}{\sigma^2} - \frac{\mathbf{1}\mathbf{1}^\top}{\sigma^2(\sigma^2+b)}\big)\mathbf{1}\right)\\
&= \frac{\kappa_0 \sigma^2}{\sigma^2 + b}\\
&= \mathcal{O}\left(\sigma^2 b^{-1}\right).
\end{aligned}$$

$\square$

**Lemma A.12.** *Suppose Assumptions 5.1 and 5.2 hold, and $Q_{\rho_k}(\boldsymbol{x})$ be defined as Eq. (11). Then, the sequence $\boldsymbol{x}_k$ generated by Algorithm 1 satisfies:*

$$\left\|\nabla Q_{\rho_k}(\boldsymbol{x}_k) - \hat{\nabla} Q_{\rho_k}(\boldsymbol{x}_k)\right\|^2 \le 3\beta_k E_{d,k,\sigma}(b_k^{(2)}) + 3m\rho_k^2 \tilde{B}_{k,m}^2 \beta_k E_{d,k,\sigma}(b_k^{(2)}) + 3m\rho_k^2 B_{\nabla c,m}^2 \beta_k e_{k,\sigma}(b_k^{(1)}). \tag{40}$$

*Proof.* By the definition of $Q_{\rho_k}$ in Eq. (11) and its surrogate $\hat{\nabla} Q_{\rho_k}$, we can write

$$\begin{aligned}
\left\|\nabla Q_{\rho_k}(\boldsymbol{x}_k) - \hat{\nabla} Q_{\rho_k}(\boldsymbol{x}_k)\right\|^2 &= \Big\|\nabla f(\boldsymbol{x}_k) - \nabla\mu_{\mathcal{D}_k}^{(f)}(\boldsymbol{x}_k) + \rho_k\big(\nabla c(\boldsymbol{x}_k)^\top c(\boldsymbol{x}_k) - \nabla c(\boldsymbol{x}_k)^\top \mu_{\mathcal{D}_k}^{(c)}(\boldsymbol{x}_k)\big)\\
&\qquad + \rho_k\big(\nabla c(\boldsymbol{x}_k)^\top \mu_{\mathcal{D}_k}^{(c)}(\boldsymbol{x}_k) - \nabla\mu_{\mathcal{D}_k}^{(c)}(\boldsymbol{x}_k)^\top \mu_{\mathcal{D}_k}^{(c)}(\boldsymbol{x}_k)\big)\Big\|^2\\
&\overset{①}{\le} 3\big\|\nabla f(\boldsymbol{x}_k) - \nabla\mu_{\mathcal{D}_k}^{(f)}(\boldsymbol{x}_k)\big\|^2 + 3\rho_k^2\big\|\nabla c(\boldsymbol{x}_k)^\top\big(c(\boldsymbol{x}_k) - \mu_{\mathcal{D}_k}^{(c)}(\boldsymbol{x}_k)\big)\big\|^2\\
&\qquad + 3\rho_k^2\big\|\big(\nabla c(\boldsymbol{x}_k) - \nabla\mu_{\mathcal{D}_k}^{(c)}(\boldsymbol{x}_k)\big)^\top \mu_{\mathcal{D}_k}^{(c)}(\boldsymbol{x}_k)\big\|^2\\
&\overset{②}{\le} 3\big\|\nabla f(\boldsymbol{x}_k) - \nabla\mu_{\mathcal{D}_k}^{(f)}(\boldsymbol{x}_k)\big\|^2 + 3\rho_k^2\big\|\nabla c(\boldsymbol{x}_k)\big\|^2\big\|c(\boldsymbol{x}_k) - \mu_{\mathcal{D}_k}^{(c)}(\boldsymbol{x}_k)\big\|^2\\
&\qquad + 3\rho_k^2\big\|\nabla c(\boldsymbol{x}_k) - \nabla\mu_{\mathcal{D}_k}^{(c)}(\boldsymbol{x}_k)\big\|^2\big\|\mu_{\mathcal{D}_k}^{(c)}(\boldsymbol{x}_k)\big\|^2\\
&\overset{③}{\le} 3\beta_k E_{d,k,\sigma}(b_k^{(2)}) + 3m\rho_k^2 \tilde{B}_{k,m}^2 \beta_k E_{d,k,\sigma}(b_k^{(2)}) + 3m\rho_k^2 B_{\nabla c,m}^2 \beta_k e_{k,\sigma}(b_k^{(1)}),
\end{aligned}$$

In the derivation above:

① utilizes the inequality: $\|\boldsymbol{x} + \boldsymbol{y} + \boldsymbol{z}\|^2 \le 3(\|\boldsymbol{x}\|^2 + \|\boldsymbol{y}\|^2 + \|\boldsymbol{z}\|^2).$

② leverages the property of the spectral norm, i.e., $\|\boldsymbol{A}\boldsymbol{x}\| \le \|\boldsymbol{A}\|\|\boldsymbol{x}\|.$

③ follows from the application of Corollary A.5, Corollary A.6 and Lemma A.8.

$\square$

## A.2. Technical Lemmas

**Lemma A.13.** *Suppose Assumptions 5.1 to 5.3 hold, and the parameters in Algorithm 1 are selected according to Eq. (17). Then, the sequence $\boldsymbol{x}_k$ generated by Algorithm 1 satisfies:*

$$d^2\left(\nabla f(\boldsymbol{x}_{k+1}) + \rho_k \nabla c(\boldsymbol{x}_{k+1})^\top c(\boldsymbol{x}_{k+1}), -N_{\mathcal{X}}(\boldsymbol{x}_{k+1})\right) \le 6\eta_k^{-2}\|\boldsymbol{x}_{k+1} - \boldsymbol{x}_k\|^2 + 3\|\nabla Q_{\rho_k}(\boldsymbol{x}_k) - \hat{\nabla} Q_{\rho_k}(\boldsymbol{x}_k)\|^2. \tag{41}$$

*Proof.* By the optimality condition of the projection step, we have:

$$0 \in \boldsymbol{x}_{k+1} - \boldsymbol{x}_k + \eta_k \hat{\nabla} Q_{\rho_k}(\boldsymbol{x}_k) + N_{\mathcal{X}}(\boldsymbol{x}_{k+1}).$$

Rearranging the terms implies:

$$\frac{1}{\eta_k}(\boldsymbol{x}_k - \boldsymbol{x}_{k+1}) + \nabla Q_{\rho_k}(\boldsymbol{x}_k) - \hat{\nabla} Q_{\rho_k}(\boldsymbol{x}_k) + \nabla Q_{\rho_k}(\boldsymbol{x}_{k+1}) - \nabla Q_{\rho_k}(\boldsymbol{x}_k) \in \nabla Q_{\rho_k}(\boldsymbol{x}_{k+1}) + N_{\mathcal{X}}(\boldsymbol{x}_{k+1}).$$

Therefore, considering the squared distance to the normal cone:

$$
\begin{aligned}
& \mathrm{d}^2\left(\nabla f(\boldsymbol{x}_{k+1}) + \rho_k \nabla c(\boldsymbol{x}_{k+1})^\top c(\boldsymbol{x}_{k+1}), -N_{\mathcal{X}}(\boldsymbol{x}_{k+1})\right) \\
=\ & \mathrm{d}^2\left(\nabla Q_{\rho_k}(\boldsymbol{x}_{k+1}), -N_{\mathcal{X}}(\boldsymbol{x}_{k+1})\right) \\
\overset{①}{\leq}\ & \left\|\frac{1}{\eta_k}(\boldsymbol{x}_k - \boldsymbol{x}_{k+1}) + \nabla Q_{\rho_k}(\boldsymbol{x}_k) - \hat{\nabla} Q_{\rho_k}(\boldsymbol{x}_k) + \nabla Q_{\rho_k}(\boldsymbol{x}_{k+1}) - \nabla Q_{\rho_k}(\boldsymbol{x}_k)\right\|^2 \\
\overset{②}{\leq}\ & 3\left[\eta_k^{-2}\|\boldsymbol{x}_k - \boldsymbol{x}_{k+1}\|^2 + \|\nabla Q_{\rho_k}(\boldsymbol{x}_k) - \hat{\nabla} Q_{\rho_k}(\boldsymbol{x}_k)\|^2 + \|\nabla Q_{\rho_k}(\boldsymbol{x}_{k+1}) - \nabla Q_{\rho_k}(\boldsymbol{x}_k)\|^2\right] \\
\overset{③}{\leq}\ & (3\eta_k^{-2} + 3L_{\rho_k,m}^2)\|\boldsymbol{x}_{k+1} - \boldsymbol{x}_k\|^2 + 3\|\nabla Q_{\rho_k}(\boldsymbol{x}_k) - \hat{\nabla} Q_{\rho_k}(\boldsymbol{x}_k)\|^2 \\
\overset{④}{\leq}\ & 6\eta_k^{-2}\|\boldsymbol{x}_{k+1} - \boldsymbol{x}_k\|^2 + 3\|\nabla Q_{\rho_k}(\boldsymbol{x}_k) - \hat{\nabla} Q_{\rho_k}(\boldsymbol{x}_k)\|^2.
\end{aligned}
$$

In the derivation above:

① follows from the definition of the distance function $\mathrm{d}(\cdot, \cdot)$.

② utilizes the inequality $\|\boldsymbol{x} + \boldsymbol{y} + \boldsymbol{z}\|^2 \leq 3(\|\boldsymbol{x}\|^2 + \|\boldsymbol{y}\|^2 + \|\boldsymbol{z}\|^2)$.

③ is derived from the smoothness property of $Q_{\rho_k}(\boldsymbol{x})$.

④ follows from the definitions of $\eta_k$ and $\rho_k$, which satisfy $\eta_k \leq L_{\rho_k,m}^{-1}$.

$\square$

**Lemma A.14.** *Suppose Assumptions 5.1 to 5.3 hold, and the parameters in Algorithm 1 are selected according to Eq. (17). Then, the sequence $\boldsymbol{x}_k$ generated by Algorithm 1 satisfies:*

$$Q_{\rho_{k+1}}(\boldsymbol{x}_{k+1}) - Q_{\rho_k}(\boldsymbol{x}_{k+1}) \leq \frac{\rho_{k+1} - \rho_k}{\gamma^2 \rho_k^2} B_{\nabla f}^2 + \frac{\rho_{k+1} - \rho_k}{\gamma^2 \rho_k^2} \mathrm{d}^2\left(\nabla f(\boldsymbol{x}_{k+1}) + \rho_k \nabla c(\boldsymbol{x}_{k+1})^\top c(\boldsymbol{x}_{k+1}), -N_{\mathcal{X}}(\boldsymbol{x}_{k+1})\right).$$
(42)

*Proof.* By the definition of the penalty function, we have:

$$
\begin{aligned}
Q_{\rho_{k+1}}(\boldsymbol{x}_{k+1}) - Q_{\rho_k}(\boldsymbol{x}_{k+1}) &= \frac{1}{2}(\rho_{k+1} - \rho_k)\|c(\boldsymbol{x}_{k+1})\|^2 \\
&\overset{①}{\leq} \frac{\rho_{k+1} - \rho_k}{2\gamma^2} \mathrm{d}^2\left(\nabla c(\boldsymbol{x}_{k+1})^\top c(\boldsymbol{x}_{k+1}), -N_{\mathcal{X}}(\boldsymbol{x}_{k+1})\right) \\
&= \frac{\rho_{k+1} - \rho_k}{2\gamma^2 \rho_k^2} \mathrm{d}^2\left(\rho_k \nabla c(\boldsymbol{x}_{k+1})^\top c(\boldsymbol{x}_{k+1}), -N_{\mathcal{X}}(\boldsymbol{x}_{k+1})\right) \\
&\overset{②}{\leq} \frac{\rho_{k+1} - \rho_k}{2\gamma^2 \rho_k^2} \left(\|\nabla f(\boldsymbol{x}_{k+1})\| + \mathrm{d}\left(\nabla f(\boldsymbol{x}_{k+1}) + \rho_k \nabla c(\boldsymbol{x}_{k+1})^\top c(\boldsymbol{x}_{k+1}), -N_{\mathcal{X}}(\boldsymbol{x}_{k+1})\right)\right)^2 \\
&\overset{③}{\leq} \frac{\rho_{k+1} - \rho_k}{\gamma^2 \rho_k^2} B_{\nabla f}^2 + \frac{\rho_{k+1} - \rho_k}{\gamma^2 \rho_k^2} \mathrm{d}^2\left(\nabla f(\boldsymbol{x}_{k+1}) + \rho_k \nabla c(\boldsymbol{x}_{k+1})^\top c(\boldsymbol{x}_{k+1}), -N_{\mathcal{X}}(\boldsymbol{x}_{k+1})\right).
\end{aligned}
$$

In the derivation above:

① follows from Assumption 5.3, which implies $\|c(\boldsymbol{x})\| \leq \frac{1}{\gamma}\mathrm{d}(\nabla c(\boldsymbol{x})^\top c(\boldsymbol{x}), -N_{\mathcal{X}}(\boldsymbol{x}))$.

② utilizes the triangle inequality for the distance to a convex set. Specifically, for any vectors $\boldsymbol{u}, \boldsymbol{v}$ and a convex set $C$:

$$\mathrm{d}(\boldsymbol{u}, C) = \inf_{\boldsymbol{z} \in C} \|\boldsymbol{u} - \boldsymbol{z}\| \leq \inf_{\boldsymbol{z} \in C} (\|\boldsymbol{u} - \boldsymbol{v}\| + \|\boldsymbol{v} - \boldsymbol{z}\|) = \|\boldsymbol{u} - \boldsymbol{v}\| + \mathrm{d}(\boldsymbol{v}, C).$$

Here, we set $\boldsymbol{u} = \rho_k \nabla c(\boldsymbol{x}_{k+1})^\top c(\boldsymbol{x}_{k+1})$ and $\boldsymbol{v} = \nabla f(\boldsymbol{x}_{k+1}) + \rho_k \nabla c(\boldsymbol{x}_{k+1})^\top c(\boldsymbol{x}_{k+1})$.

③ combines the inequality $\|\boldsymbol{x} + \boldsymbol{y}\|^2 \leq 2(\|\boldsymbol{x}\|^2 + \|\boldsymbol{y}\|^2)$ with the boundedness of the gradient, i.e., $\|\nabla f(\boldsymbol{x})\| \leq B_{\nabla f}$.

$\square$

**Lemma A.15.** *Suppose Assumptions 5.1 to 5.3 hold, and the parameters in Algorithm 1 are selected according to Eq. (17). Then, the sequence $\boldsymbol{x}_k$ generated by Algorithm 1 satisfies:*

$$Q_{\rho_k}(\boldsymbol{x}_{k+1}) - Q_{\rho_k}(\boldsymbol{x}_k) \leq \frac{\eta_k}{2} \|\nabla Q_{\rho_k}(\boldsymbol{x}_k) - \hat{\nabla} Q_{\rho_k}(\boldsymbol{x}_k)\|^2 - \frac{1}{4\eta_k} \|\boldsymbol{x}_{k+1} - \boldsymbol{x}_k\|^2. \tag{43}$$

*Proof.* By the optimality of the projection step, we have:

$$\langle \boldsymbol{x}_{k+1} - \boldsymbol{x}_k + \eta_k \hat{\nabla} Q_{\rho_k}(\boldsymbol{x}_k), \boldsymbol{x}_k - \boldsymbol{x}_{k+1} \rangle \geq 0.$$

Rearranging this inequality implies:

$$\langle \hat{\nabla} Q_{\rho_k}(\boldsymbol{x}_k), \boldsymbol{x}_{k+1} - \boldsymbol{x}_k \rangle \leq -\frac{1}{\eta_k} \|\boldsymbol{x}_{k+1} - \boldsymbol{x}_k\|^2. \tag{44}$$

Now, consider the expansion of the penalty function. We have:

$$Q_{\rho_k}(\boldsymbol{x}_{k+1}) \overset{①}{\leq} Q_{\rho_k}(\boldsymbol{x}_k) + \langle \hat{\nabla} Q_{\rho_k}(\boldsymbol{x}_k), \boldsymbol{x}_{k+1} - \boldsymbol{x}_k \rangle + \langle \nabla Q_{\rho_k}(\boldsymbol{x}_k) - \hat{\nabla} Q_{\rho_k}(\boldsymbol{x}_k), \boldsymbol{x}_{k+1} - \boldsymbol{x}_k \rangle + \frac{L_{\rho_k, m}}{2} \|\boldsymbol{x}_{k+1} - \boldsymbol{x}_k\|^2$$

$$\overset{②}{\leq} Q_{\rho_k}(\boldsymbol{x}_k) + \langle \nabla Q_{\rho_k}(\boldsymbol{x}_k) - \hat{\nabla} Q_{\rho_k}(\boldsymbol{x}_k), \boldsymbol{x}_{k+1} - \boldsymbol{x}_k \rangle + \left( \frac{L_{\rho_k, m}}{2} - \frac{1}{\eta_k} \right) \|\boldsymbol{x}_{k+1} - \boldsymbol{x}_k\|^2$$

$$\overset{③}{\leq} Q_{\rho_k}(\boldsymbol{x}_k) + \frac{\eta_k}{2} \|\nabla Q_{\rho_k}(\boldsymbol{x}_k) - \hat{\nabla} Q_{\rho_k}(\boldsymbol{x}_k)\|^2 + \left( -\frac{1}{2\eta_k} + \frac{L_{\rho_k, m}}{2} \right) \|\boldsymbol{x}_{k+1} - \boldsymbol{x}_k\|^2$$

$$\overset{④}{\leq} Q_{\rho_k}(\boldsymbol{x}_k) + \frac{\eta_k}{2} \|\nabla Q_{\rho_k}(\boldsymbol{x}_k) - \hat{\nabla} Q_{\rho_k}(\boldsymbol{x}_k)\|^2 - \frac{1}{4\eta_k} \|\boldsymbol{x}_{k+1} - \boldsymbol{x}_k\|^2.$$

In the derivation above:

① follows from the property that $Q_{\rho_k}(\boldsymbol{x})$ is $L_{\rho_k, m}$-smooth. Note that the linear term $\langle \nabla Q_{\rho_k}(\boldsymbol{x}_k), \boldsymbol{x}_{k+1} - \boldsymbol{x}_k \rangle$ is decomposed into $\langle \hat{\nabla} Q, \dots \rangle + \langle \nabla Q - \hat{\nabla} Q, \dots \rangle$.

② obtained by directly substituting the projection inequality (44) into the expression.

③ utilizes Young's inequality in the form $\langle \boldsymbol{x}, \boldsymbol{y} \rangle \leq \frac{\eta_k \|\boldsymbol{x}\|^2}{2} + \frac{\|\boldsymbol{y}\|^2}{2\eta_k}$.

④ holds because our parameter selection satisfies the condition $\eta_k \leq (2L_{\rho_k, m})^{-1}$.

$\square$

**Lemma A.16.** *Suppose Assumptions 5.1 to 5.3 hold, and the parameters in Algorithm 1 are selected according to Eq. (17). Let $\{\boldsymbol{x}_k\}$ be the sequence generated by Algorithm 1 and $\tilde{K}_m$ be a sufficiently large number (specifically, $\tilde{K}_m = 70\gamma^{-\frac{8}{3}} L_m^{\frac{4}{3}}$). For any $k > \tilde{K}_m$, the following inequality holds:*

$$\frac{\eta_k}{48} d^2 \left( \nabla f(\boldsymbol{x}_{k+1}) + \rho_k \nabla c(\boldsymbol{x}_{k+1})^\top c(\boldsymbol{x}_{k+1}), -N_{\mathcal{X}}(\boldsymbol{x}_{k+1}) \right)$$

$$\leq Q_{\rho_k}(\boldsymbol{x}_k) - Q_{\rho_{k+1}}(\boldsymbol{x}_{k+1}) + \frac{5\eta_k}{8} \|\nabla Q_{\rho_k}(\boldsymbol{x}_k) - \hat{\nabla} Q_{\rho_k}(\boldsymbol{x}_k)\|^2 + \frac{(\rho_{k+1} - \rho_k) B_{\nabla f}^2}{\gamma^2 \rho_k^2}. \tag{45}$$

*Proof.* The proof proceeds by combining previous lemmas and analyzing the parameter selection for large $k$. Based on the properties of Algorithm 1, we derive the bound as follows:

$$\frac{\eta_k}{24}\mathrm{d}^2\left(\nabla f(\boldsymbol{x}_{k+1}) + \rho_k\nabla c(\boldsymbol{x}_{k+1})^\top c(\boldsymbol{x}_{k+1}), -N_{\mathcal{X}}(\boldsymbol{x}_{k+1})\right)$$

$$\overset{①}{\leq} \frac{1}{4\eta_k}\|\boldsymbol{x}_{k+1} - \boldsymbol{x}_k\|^2 - \frac{\eta_k}{2}\|\nabla Q_{\rho_k}(\boldsymbol{x}_k) - \hat{\nabla} Q_{\rho_k}(\boldsymbol{x}_k)\|^2 + \frac{5\eta_k}{8}\|\nabla Q_{\rho_k}(\boldsymbol{x}_k) - \hat{\nabla} Q_{\rho_k}(\boldsymbol{x}_k)\|^2$$

$$\overset{②}{\leq} Q_{\rho_k}(\boldsymbol{x}_k) - Q_{\rho_{k+1}}(\boldsymbol{x}_{k+1}) + Q_{\rho_{k+1}}(\boldsymbol{x}_{k+1}) - Q_{\rho_k}(\boldsymbol{x}_{k+1}) + \frac{5\eta_k}{8}\|\nabla Q_{\rho_k}(\boldsymbol{x}_k) - \hat{\nabla} Q_{\rho_k}(\boldsymbol{x}_k)\|^2 \qquad (46)$$

$$\overset{③}{\leq} Q_{\rho_k}(\boldsymbol{x}_k) - Q_{\rho_{k+1}}(\boldsymbol{x}_{k+1}) + \frac{\rho_{k+1} - \rho_k}{\gamma^2\rho_k^2}\mathrm{d}^2\left(\nabla f(\boldsymbol{x}_{k+1}) + \rho_k\nabla c(\boldsymbol{x}_{k+1})^T c(\boldsymbol{x}_{k+1}), -N_{\mathcal{X}}(\boldsymbol{x}_{k+1})\right)$$

$$+ \frac{\rho_{k+1} - \rho_k}{\gamma^2\rho_k^2}B_{\nabla f}^2 + \frac{5\eta_k}{8}\|\nabla Q_{\rho_k}(\boldsymbol{x}_k) - \hat{\nabla} Q_{\rho_k}(\boldsymbol{x}_k)\|^2.$$

The derivation above follows from Lemma A.13, Lemma A.15 and Lemma A.14, respectively.

Furthermore, when $k > \tilde{K}_m$, we examine the coefficient term associated with the stationarity residual:

$$\frac{\rho_{k+1} - \rho_k}{\gamma^2\rho_k^2} < \frac{\frac{1}{4}k^{-\frac{3}{4}}}{\gamma^2 k^{\frac{1}{2}}} < \frac{1}{4}\gamma^{-2}\tilde{K}_m^{-\frac{3}{4}}k^{-\frac{1}{2}} < \frac{1}{96L_m}k^{-\frac{1}{2}} = \frac{\eta_k}{48}$$

By rearranging the terms in Eq. (46), we obtain:

$$\frac{\eta_k}{48}\mathrm{d}^2(\cdot) \leq \left(\frac{\eta_k}{24} - \frac{\rho_{k+1} - \rho_k}{\gamma^2\rho_k^2}\right)\mathrm{d}^2(\cdot) \leq Q_{\rho_k}(\boldsymbol{x}_k) - Q_{\rho_{k+1}}(\boldsymbol{x}_{k+1}) + \dots,$$

which completes the proof. $\qquad\square$

### A.3. Proof of Theorem 5.5

*Proof.*

$$\sum_{k=1}^{K}\frac{\eta_k}{48}\mathrm{d}^2\left(\nabla f(\boldsymbol{x}_{k+1}) + \rho_k\nabla c(\boldsymbol{x}_{k+1})^\top c(\boldsymbol{x}_{k+1}), -N_{\mathcal{X}}(\boldsymbol{x}_{k+1})\right)$$

$$\leq \sum_{k=1}^{\tilde{K}_m - 1}\frac{\eta_k}{24}\mathrm{d}^2\left(\nabla f(\boldsymbol{x}_{k+1}) + \rho_k\nabla c(\boldsymbol{x}_{k+1})^\top c(\boldsymbol{x}_{k+1}), -N_{\mathcal{X}}(\boldsymbol{x}_{k+1})\right)$$

$$+ \sum_{k=\tilde{K}_m}^{K}\frac{\eta_k}{48}\mathrm{d}^2\left(\nabla f(\boldsymbol{x}_{k+1}) + \rho_k\nabla c(\boldsymbol{x}_{k+1})^\top c(\boldsymbol{x}_{k+1}), -N_{\mathcal{X}}(\boldsymbol{x}_{k+1})\right)$$

$$\overset{①}{\leq} Q_1(\boldsymbol{x}_1) - Q_{\tilde{K}_m}(\boldsymbol{x}_{\tilde{K}_m}) + \frac{1}{2}(\rho_{\tilde{K}_m} - \rho_1)B_{c,m}^2 + \sum_{k=1}^{\tilde{K}_m - 1}\frac{5\eta_k}{8}\|\nabla Q_{\rho_k}(\boldsymbol{x}_k) - \hat{\nabla} Q_{\rho_k}(\boldsymbol{x}_k)\|^2$$

$$+ Q_{\tilde{K}_m}(\boldsymbol{x}_{\tilde{K}_m}) - Q_{\rho_{K+1}}(\boldsymbol{x}_{K+1}) + \sum_{k=\tilde{K}_m}^{K}\frac{(\rho_{k+1} - \rho_k)B_{\nabla f}^2}{\gamma^2\rho_k^2} + \sum_{k=\tilde{K}_m}^{K}\frac{5\eta_k}{8}\|\nabla Q_{\rho_k}(\boldsymbol{x}_k) - \hat{\nabla} Q_{\rho_k}(\boldsymbol{x}_k)\|^2$$

$$\overset{②}{\leq} Q_1(\boldsymbol{x}_1) + B_f + \frac{1}{2}(\rho_{\tilde{K}_m} - \rho_1)B_{c,m}^2 + \sum_{k=1}^{K}\frac{(\rho_{k+1} - \rho_k)B_{\nabla f}^2}{\gamma^2\rho_k^2} + \sum_{k=1}^{K}\frac{5\eta_k}{8}\|\nabla Q_{\rho_k}(\boldsymbol{x}_k) - \hat{\nabla} Q_{\rho_k}(\boldsymbol{x}_k)\|^2$$

The justification of the steps are as follows:

① is obtained using Eq. (46) (step ②) and Lemma A.16.

② utilizes the lower bound $Q_\rho(\boldsymbol{x}) \geq -B_f$ for any $\rho \geq 0$ and the non-negativity of $(\rho_{k+1} - \rho_k)B_{\nabla f}^2/(\gamma^2 \rho_k^2)$.

Furthermore, given that $\eta_k \geq \eta_K$ for any $k \leq K$, we have:

$$
\frac{1}{K}\sum_{k=1}^{K} \mathrm{d}^2\big(\nabla f(\boldsymbol{x}_{k+1}) + \rho_k \nabla c(\boldsymbol{x}_{k+1})^\top c(\boldsymbol{x}_{k+1}), -N_{\mathcal{X}}(\boldsymbol{x}_{k+1})\big)
$$

$$
\leq \frac{1}{K\eta_K}\sum_{k=1}^{K} \eta_k \mathrm{d}^2\big(\nabla f(\boldsymbol{x}_{k+1}) + \rho_k \nabla c(\boldsymbol{x}_{k+1})^\top c(\boldsymbol{x}_{k+1}), -N_{\mathcal{X}}(\boldsymbol{x}_{k+1})\big) \tag{47}
$$

$$
\leq \frac{48}{K\eta_K}\Big(Q_1(\boldsymbol{x}_1) + B_f + \frac{1}{2}(\rho_{\tilde{K}_m} - \rho_1)B_{c,m}^2 + \sum_{k=1}^{K}\frac{(\rho_{k+1} - \rho_k)B_{\nabla f}^2}{\gamma^2 \rho_k^2} + \sum_{k=1}^{K}\frac{5\eta_k}{8}\|\nabla Q_{\rho_k}(\boldsymbol{x}_k) - \hat{\nabla}Q_{\rho_k}(\boldsymbol{x}_k)\|^2\Big)
$$

Note that with the choice $\rho_k = k^{1/4}$, the term $\rho_k^{-2}(\rho_{k+1} - \rho_k)$ scales as $\mathcal{O}(k^{-5/4})$. Since the series $\sum_{k=1}^{\infty} k^{-5/4}$ converges, the cumulative sum is bounded by a finite constant $S_\rho < +\infty$. Regarding the gradient error term, by substituting the expanded form of Eq. (40), we further derive:

$$
\frac{1}{K}\sum_{k=1}^{K} \mathrm{d}^2\big(\nabla f(\boldsymbol{x}_{k+1}) + \rho_k \nabla c(\boldsymbol{x}_{k+1})^\top c(\boldsymbol{x}_{k+1}), -N_{\mathcal{X}}(\boldsymbol{x}_{k+1})\big)
$$

$$
\leq \frac{1}{\sqrt{K}}\Big(96L_m\big(Q_1(\boldsymbol{x}_1) + B_f + \frac{1}{2}(\rho_{\tilde{K}_m} - \rho_1)B_{c,m}^2 + \frac{S_\rho B_{\nabla f}^2}{\gamma^2}\big) + 90\beta_k\big(B_{\nabla c,m}^2 m e_{k,\sigma}(b_k^{(1)}) + (m\tilde{B}_{k,m}^2 + 2L_m\eta_k)E_{d,k,\sigma}(b_k^{(2)})\big)\Big)
$$

$$
\leq \frac{1}{\sqrt{K}}(C_{1,m} + C_2\mathcal{E}_K) \tag{48}
$$

Here, $C_{1,m} = 96L_m\big(Q_1(\boldsymbol{x}_1) + B_f + \frac{1}{2}(\rho_{\tilde{K}_m} - \rho_1)B_{c,m}^2 + S_\rho B_{\nabla f}^2/\gamma^2\big) = \mathcal{O}(m^{\frac{7}{3}}), C_2 = 90$ and $\mathcal{E}_K = \sum_{k=1}^{K}\beta_k\big(c_{1,m}e_{k,\sigma}(b_k^{(1)}) + (c_{2,m}\tilde{B}_{k,m}^2 + c_{3,m}\eta_k)E_{d,k,\sigma}(b_k^{(2)})\big)$, where $c_{1,m} = mB_{\nabla c,m}^2 = \mathcal{O}(m^2), c_{2,m} = m = \mathcal{O}(m), c_{3,m} = 2L_m = \mathcal{O}(m)$.

For constraint residual we have:

$$
\frac{1}{K}\sum_{k=1}^{K}\|c(\boldsymbol{x}_{k+1})\|^2 \overset{①}{\leq} \frac{2}{\gamma^2 K}\sum_{k=1}^{K}\Big(\frac{B_{\nabla f}^2}{\rho_k^2} + \frac{1}{\rho_k^2}\mathrm{d}^2\big(\nabla f(\boldsymbol{x}_{k+1}) + \rho_k \nabla c(\boldsymbol{x}_{k+1})^\top c(\boldsymbol{x}_{k+1}), -N_{\mathcal{X}}(\boldsymbol{x}_{k+1})\big)\Big)
$$

$$
\overset{②}{\leq} \frac{2}{\gamma^2 K}\big(2B_{\nabla f}^2\sqrt{K} + C_{1,m} + C_2\mathcal{E}_K\big)
$$

$$
\overset{③}{=} \frac{C_3}{\sqrt{K}} + \frac{C_{4,m} + C_5\mathcal{E}_K}{K}
$$

In the derivation above:

① follows from Lemma A.14.

② is derived by by applying the series inequality $\sum_{k=1}^{K} k^{-1/2} \leq 2\sqrt{K}$ and combine Eq. (47) and Eq. (48), which implies

$$
\sum_{k=1}^{K}\frac{1}{\sqrt{k}}\mathrm{d}^2\big(\nabla f(\boldsymbol{x}_{k+1}) + \rho_k \nabla c(\boldsymbol{x}_{k+1})^\top c(\boldsymbol{x}_{k+1}), -N_{\mathcal{X}}(\boldsymbol{x}_{k+1})\big) \leq C_{1,m} + C_2\mathcal{E}_K
$$

③ is obtained by setting $C_3 = 4B_{\nabla f}^2\gamma^{-2}, C_{4,m} = 2C_{1,m}\gamma^{-2} = \mathcal{O}(m^{\frac{7}{3}})$ and $C_5 = 2C_2\gamma^{-2}$.

Finally, the proof is completed by setting $\lambda_{k+1} = \rho_k c(\boldsymbol{x}_{k+1})$. $\qquad\square$

## A.4. Proof of Corollary 5.6

*Proof.* The proof of Corollary 5.6 primarily focuses on analyzing the orders of the following two quantities.:

$$e = \sum_{k=1}^{K} c_{1,m} \beta_k e_{d,k}(b_k^{(1)})$$

$$E = \sum_{k=1}^{K} c_{2,m} \beta_k \tilde{B}_{k,m}^2 E_{d,k,\sigma}(b_k^{(2)})$$

From Lemma A.10 and Lemma A.11, we can bound $E$ and $e$ as:

$$e \lesssim \sigma^2 m^2 \sum_{k=1}^{K} b_k^{(1)^{-1}} \log k,$$

$$E \lesssim \sigma m^2 d^{\frac{3}{2}} \sum_{k=1}^{K} b_k^{(2)^{-\frac{1}{2}}} \log^2 k.$$

When $b_k^{(2)} = dk^2$, the term $\sum_{k=1}^{K} b_k^{(2)^{-\frac{1}{2}}} \log^2 k$ has a cumulative growth rate of $\mathcal{O}(d^{-\frac{1}{2}} \log^3 K)$. Meanwhile, the term $\sum_{k=1}^{K} b_k^{(1)^{-1}} \log k$ is bounded by $\mathcal{O}(\log^2 K)$ when $b_k^{(1)}$ is set to $k$. The total sample size is given by $n = (m+1) \sum_{k=1}^{K} (dk^2 + k) = \mathcal{O}(mdK^3)$. And thus $K = \mathcal{O}(m^{-\frac{1}{3}} d^{-\frac{1}{3}} n^{\frac{1}{3}})$. Finally, combining these results:

$$\frac{1}{K} \sum_{k=1}^{K} \mathrm{d}^2(\nabla f(\boldsymbol{x}_{k+1}) + \nabla c(\boldsymbol{x}_{k+1})\lambda_{k+1}, -N_{\mathcal{X}}(\boldsymbol{x}_{k+1}))$$

$$= \mathcal{O}((E + e + C_{1,m})K^{-\frac{1}{2}})$$

$$= \mathcal{O}(m^{\frac{7}{3}} K^{-\frac{1}{2}}) + \tilde{\mathcal{O}}(\sigma m^2 d K^{-\frac{1}{2}})$$

$$= \mathcal{O}(m^{\frac{5}{2}} d^{\frac{1}{6}} n^{-\frac{1}{6}}) + \tilde{\mathcal{O}}(\sigma m^{\frac{13}{6}} d^{\frac{7}{6}} n^{-\frac{1}{6}}).$$

$$\frac{1}{K} \sum_{k=1}^{K} \|c(\boldsymbol{x}_{k+1})\|^2 = \mathcal{O}(K^{-\frac{1}{2}}) + \mathcal{O}((E + e + C_{1,m})K^{-1})$$

$$= \mathcal{O}(K^{-\frac{1}{2}}) + \mathcal{O}(m^{\frac{7}{3}} K^{-1}) + \tilde{\mathcal{O}}(\sigma m^2 d K^{-1})$$

$$= \mathcal{O}(m^{\frac{1}{6}} d^{\frac{1}{6}} n^{-\frac{1}{6}}) + \mathcal{O}(m^{\frac{8}{3}} d^{\frac{1}{3}} n^{-\frac{1}{3}}) + \tilde{\mathcal{O}}(\sigma m^{\frac{7}{3}} d^{\frac{4}{3}} n^{-\frac{1}{3}})$$

$\square$

## B. Comparative Analysis of Mechanisms

This section clarifies the failure modes of trust-region-based CBO methods that motivate LCBO. We do not claim that such methods cannot recover from these cases. Rather, under a limited evaluation budget, repeated shrinkage and restarts may consume many evaluations without producing feasible improvement.

**Mechanism analysis** Figure 3 illustrates a curved constraint boundary. When the trust-region center approaches such a boundary, SCBO may face a structural dilemma: candidates aligned with the local descent direction tend to violate the constraints, while feasible candidates inside the current region may provide no improvement over the incumbent. Under SCBO's success/failure-based update rule, both cases can be recorded as local failures, causing the trust-region radius to shrink.

A smaller radius further restricts exploration along the boundary and can eventually trigger a restart once the radius falls below the minimum length. Although restarts can in principle allow recovery, and the collected observations are still retained by the GP model, the shrinkage–restart process may spend a substantial portion of the budget on candidates that

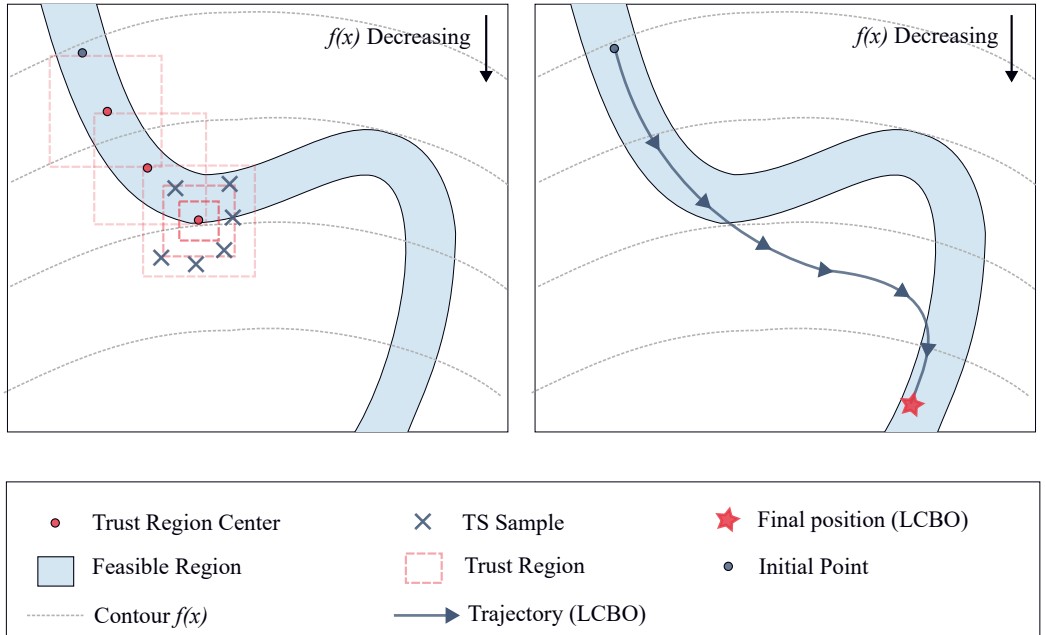

*Figure 3.* **Mechanism Comparison between SCBO and LCBO.** Near a curved constraint boundary, SCBO may face a structural dilemma: descent-aligned candidates tend to be infeasible, while feasible candidates inside the trust region may yield no improvement. Both outcomes can be counted as local failures under the success/failure-based update rule, leading to trust-region shrinkage and possible restarts. LCBO instead exploits the gradient of a quadratic-penalty surrogate, enabling progress along the boundary while correcting constraint violation.

yield no feasible improvement. LCBO addresses this failure mode by using the gradient of a quadratic-penalty surrogate, which combines objective descent with constraint-violation correction. Thus, even if an iterate temporarily moves into the infeasible region near a boundary, the penalty gradient can guide it back toward feasibility while allowing continued progress along the boundary.

**Empirical confirmation on the truss task.** On the truss design task, SCBO restarts provide direct evidence of repeated shrinkage, since a restart is triggered if and only if the trust-region radius falls below the minimum length. Across all 10 independent runs, every run experienced at least two restarts. The first restart occurred at a median of 350 evaluations, with interquartile range 338–383, and the second at 747 evaluations, with interquartile range 657–756. The timing of the first restart closely aligns with the onset of the performance plateau in Figure 2, suggesting that the geometric obstruction described above contributes to SCBO's stagnation on this task.

## C. Ablation Studies

### C.1. Ablation on Batch Schedule

**Motivation and Experimental Setup** In the theoretical analysis, our high-probability convergence bound is predicated on an increasing batch size schedule (specifically, $b_1^{(k)} \propto k, b_k^{(2)} \propto k^2$). However, in practical high-dimensional BO with finite evaluation budgets, strictly adhering to such a rapid growth schedule can lead to "budget starvation", where the algorithm exhausts the evaluation budget within very few iterations. To bridge this theory-practice gap and justify our choice of fixed mini-batches in the main experiments, we conducted a systematic ablation study. This study aims to decouple the effects of per-iteration convergence quality from sample efficiency, demonstrating why fixed mini-batches act as a superior stochastic approximation in budget-constrained settings.

We performed ablation experiments on synthetic benchmarks with varying dimensions ($d \in \{25, 50, 100\}$) to observe the impact of dimensionality. All experiments were conducted with a fixed total budget of 5,000 evaluations. We report the median and quartiles of the best feasible objective found so far, computed across 5 distinct random seeds. We compared three distinct batch size schedules:

- **Fixed Mini-batch:** A small, constant batch size ($b_k^{(1)} = 2, b_k^{(2)} = 5$) designed to facilitate more frequent updates.

- **Fixed Large-batch:** A fixed batch size scaling with dimension ($b_k^{(1)} = 5, b_k^{(2)} = d$). This represents a strategy that prioritizes spatial coverage in each step but sacrifices iteration count.

- **Growing Batch:** While theory suggests $b_k^{(1)} \propto k, b_k^{(2)} \propto k^2$, such a schedule would deplete the 5,000-evaluation budget in fewer than 30 iterations. To provide a fair but challenging baseline, we adopted a relaxed growth schedule ($b_k^{(1)} = \lfloor \log(k+1) + 1 \rfloor, b_k^{(2)} = \lfloor 0.5k + 5 \rfloor$). This serves as a "steel-manned" proxy for the theoretical setting: if our method outperforms this linearly growing baseline (which is much more budget-friendly than $\mathcal{O}(k^2)$), the conclusion holds a fortiori for the strictly theoretical $\mathcal{O}(k^2)$ schedule.

**Results and Analysis** We present the optimization progress from two complementary perspectives: **Best Feasible Objective vs. Iterations** (Figure 4) and **Best Feasible Objective vs. Evaluations** (Figure 5).

Verification of Theoretical Intuition (Best Feasible Objective vs. Iterations): When plotted against the number of iterations (updates), the Fixed Large-batch and Growing Batch strategies often achieve a steeper descent per step compared to the Mini-batch approach. This empirically validates our theoretical analysis: larger or growing batches indeed provide more accurate gradient estimates, leading to more stable and "correct" updates in the optimization landscape.

Practical Superiority (Best Feasible Objective vs. Evaluations): However, when plotted against the actual number of function evaluations (the true cost in BO), the Fixed Mini-batch strategy demonstrates superior sample efficiency, particularly within the initial 1,000 evaluations. By tolerating higher variance in gradient estimation, the Mini-batch strategy performs orders of magnitude more updates (e.g., $\approx 700$ updates vs. $< 50$ updates for $b = d$ in 100D synthetic task). This allows the algorithm to traverse the search space effectively and reach lower regret values within the fixed budget.

**Discussion: The SGD Analogy** Our findings can be interpreted through the lens of Stochastic Gradient Descent (SGD) in deep learning. Just as SGD is often preferred over full-batch gradient descent despite its noisy gradient estimates, our Fixed Mini-batch LCBO prefers frequent, noisy updates over rare, precise ones.

Beyond sample efficiency, the choice of mini-batches may offer an implicit regularization benefit. The stochasticity introduced by small batch sizes can help the optimization trajectory escape shallow local optima or saddle points, a phenomenon well-documented in non-convex optimization. While our theoretical proof relies on vanishing variance for exact convergence, empirical evidence suggests that maintaining a certain level of "exploration noise" via fixed mini-batches is not only practically necessary but potentially beneficial for global search in complex high-dimensional landscapes.

### C.2. Ablation on Local GP, Local Search Box, and Gradient Normalization

The implementation strategies described in Section 6.1, which include (i) local GPs trained on the $N_m$ most recent points, (ii) restricting acquisition function optimization to a dynamic local box $[x_k - \delta, x_k + \delta]$, and (iii) Euclidean-norm normalization of $\hat{\nabla} Q_{\rho_k}(x_k)$, are all inherited from the GIBO framework (Müller et al., 2021) and serve distinct practical purposes. The local GP reduces the training complexity from $\mathcal{O}(N^3)$ to $\mathcal{O}(N_m^3)$ and naturally adapts to local landscape changes by discarding stale data. The local search box concentrates the candidate set near the current iterate, where the gradient information is most informative for the next descent step. Gradient normalization decouples direction from magnitude, providing adaptive step-size control analogous to Adam and RMSprop.

To verify that the primary gains of LCBO arise from the core mechanism, that is, penalty-based gradient descent on GP surrogates, rather than from these auxiliary components, we ablate each one individually on the Truss (25D) and Synthetic (50D) tasks. Results are reported in Figure 6.

Across both tasks, every ablated variant remains competitive with or surpasses the strongest CBO baseline, confirming that none of these components is individually responsible for LCBO's advantage. Instead, they serve as practical engineering choices.

### C.3. Ablation on the Penalty Factor Schedule

A natural concern is whether LCBO's growing penalty schedule $\rho_k \propto k^{1/4}$ requires per-task tuning to perform well. We address this concern from two complementary angles.

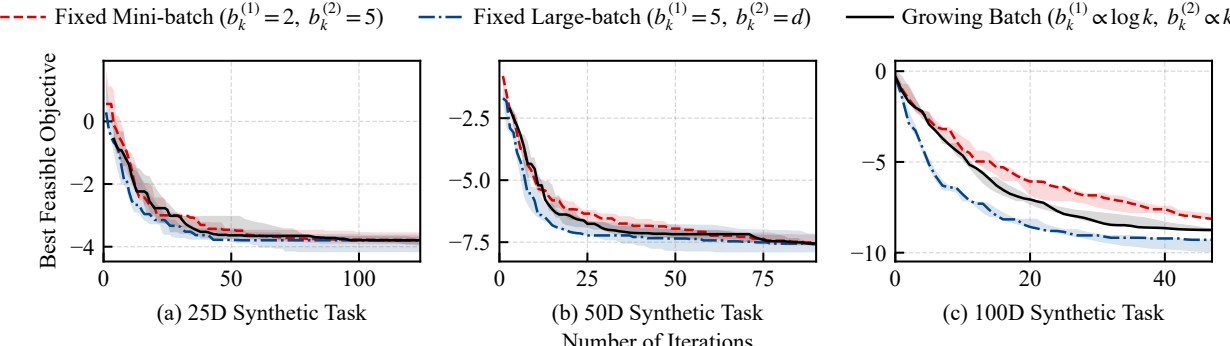

*Figure 4.* **Best Feasible Objective vs. Iterations** The x-axis represents the number of algorithm iterations, while the y-axis shows the best feasible objective value found so far (lower is better). We truncated the results to the minimum iteration count across the three batch schedules.

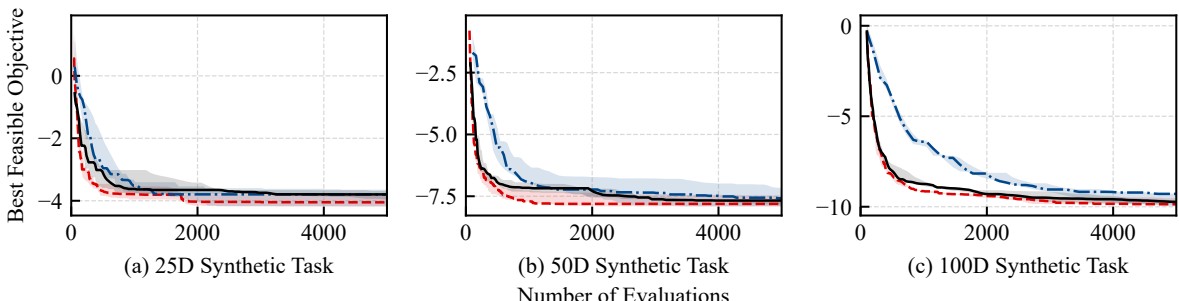

*Figure 5.* **Best Feasible Objective vs. Evaluations** The x-axis represents the number of function evaluations, while the y-axis shows the best feasible objective value found so far (lower is better).

**Practical robustness across tasks.** Our implementation applies Z-score normalization to all function outputs (objective and constraints) before GP fitting, as detailed in Appendix E. This standardization largely removes the influence of problem-specific constraint magnitudes on the effective penalty scale, which is a key reason why a single schedule works across all benchmarks in Section 6 without any task-specific adjustment.

**Ablation: growing schedule vs. fixed constants.** To directly isolate the role of the penalty design within our local search framework, we compare LCBO's growing schedule against four fixed-penalty variants $\rho_k \equiv \rho \in \{1, 10, 100, 1000\}$ on the Truss and 50D Synthetic tasks. Since all variants share the same local exploration–exploitation mechanism, any performance difference is attributable solely to the penalty strategy. Figure 7 shows the corresponding optimization curves.

The results illustrate the classical trade-off inherent in fixed-penalty methods. When $\rho$ is too small, the penalty term is insufficient to drive the iterates toward the feasible region; when $\rho$ is too large, the penalty dominates the gradient and slows progress on the objective. Both regimes are observed in our experiments. On the Truss task, where the constraints are tight, $\rho_k \equiv 1$ remains close to its initial level throughout the budget, indicating that the penalty is too weak to drive the iterates into the feasible region. On the 50D Synthetic task, $\rho_k \equiv 1000$ exhibits the slowest descent among the fixed variants, consistent with the over-penalization regime in which the constraint term overwhelms the objective signal. LCBO's growing schedule is designed to navigate this trade-off in a principled manner: moderate penalization in the early phase allows the algorithm to make progress on the objective, while the gradually increasing $\rho_k$ ensures that constraint enforcement tightens as the iterates approach feasibility. This schedule is precisely what underpins our convergence analysis in Theorem 5.5, and the ablation results show that it also matches or outperforms the best fixed constant on each task without any problem-specific tuning. The growing schedule thus combines a theoretical convergence guarantee with practical robustness in the black-box setting, where the appropriate magnitude of $\rho$ is not known a priori.

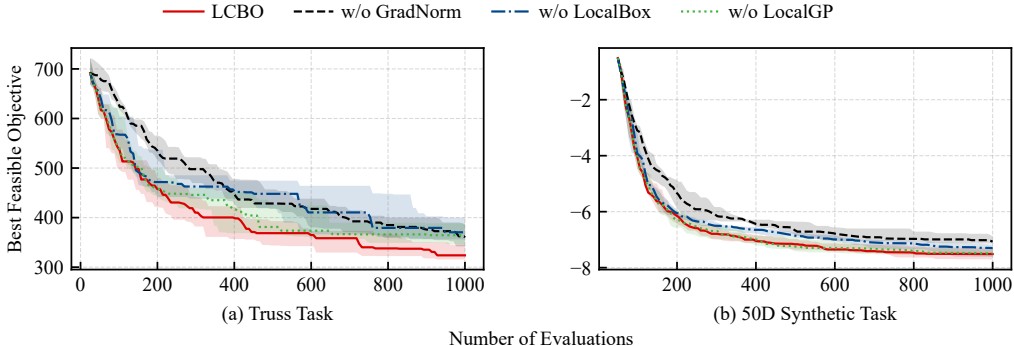

*Figure 6.* **Optimization progress of the component ablation on the Truss (25D) and 50D Synthetic tasks.** The plotting conventions and axes definitions are identical to Figure 1.

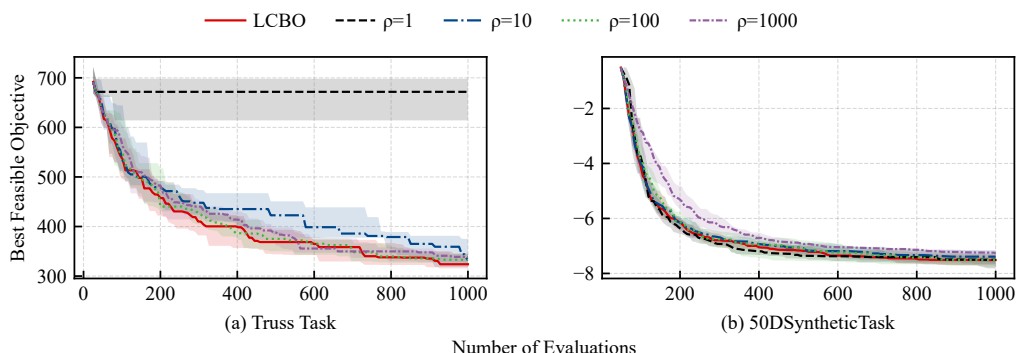

*Figure 7.* **Optimization progress of the penalty schedule ablation on the Truss (25D) and 50D Synthetic tasks.** The plotting conventions and axes definitions are identical to Figure 1.

## D. Experiment Setup

### D.1. 25-Bar Truss Design (25D)

The standard 25-bar truss design problem has been a canonical benchmark in structural optimization (Camp & Bichon, 2004). In this problem, the design variables correspond to the cross-sectional areas of the 25 truss members. The search space is constrained within the range $[0.01, 5.0]$ in$^2$, where the lower bound is explicitly imposed to prevent singularity of the stiffness matrix during finite element analysis. The structure is composed of an aluminum alloy with a Young's modulus of $E = 1.0 \times 10^7$ psi and a density of $\rho = 0.1$ lb/in$^3$. The primary optimization objective is to minimize the total weight of the truss. Under the standard loading scenario (Case 1), the design must satisfy strict physical constraints. Specifically, the absolute value of the axial stress in each member must not exceed $40,000$ psi, and the absolute displacement of all free nodes along any coordinate axis must remain within $0.35$ in. To efficiently manage these 25 stress constraints and 18 displacement constraints, we employ the LogSumExp function for constraint aggregation and smoothing. By setting the smoothing factor $\alpha = 20.0$, we transform the multidimensional constraint criteria into two aggregated functions (i.e., an aggregated stress constraint and an aggregated displacement constraint). This formulation alleviates the need to maintain numerous GPs simultaneously, thereby reducing computational complexity.

### D.2. Stepped Cantilever Beam Design (50D)

The Stepped Cantilever Beam benchmark is a standard problem in engineering design literature (e.g., (Thanedar & Vanderplaats, 1995)). The problem involves determining the optimal cross-sectional dimensions for a cantilever beam of length $L = 100$ in, which is discretized into $N = 25$ distinct segments. The design variables consist of the width ($w_i$) and height ($h_i$) for each segment, resulting in a total of 50 independent variables bounded within the range $[0.5, 5.0]$ in. to ensure geometric validity and prevent singularity in the stiffness matrix. The beam is modeled using Euler-Bernoulli

theory with solid rectangular cross-sections, assuming a Young's modulus of $E = 2.9 \times 10^7$ psi and subjected to a static vertical point load of $P = 500$ lb at the free tip. The primary objective is to minimize the total volume of the structure. The design is subject to two critical physical constraints: the vertical displacement at the beam tip must not exceed $2.5$ in, and the maximum bending stress within any segment must be strictly less than $40,000$ psi. Since the stress constraint is evaluated locally at each segment, we utilize the LogSumExp function with a smoothing parameter $\alpha = 20.0$ to aggregate these local violations into a single global stress constraint.

### D.3. MuJoCo HalfCheetah (102D)

To evaluate the optimization framework on high-dimensional control tasks, we utilize the HalfCheetah-v4 environment from the Gymnasium MuJoCo suite (Todorov et al., 2012). The problem involves optimizing a linear policy to maximize the locomotion speed of a 2D planar robot. The optimization variables represent the weights of a linear controller mapping the 17-dimensional observation space to the 6-dimensional action space, resulting in a total of 102 decision variables bounded within $[-0.2, 0.2]$. To ensure numerical stability and consistent policy behavior, we implement a static normalization scheme where input observations are standardized using fixed mean and variance statistics derived from a pre-calculation phase of 2,000 random steps, followed by a hyperbolic tangent ($\tanh$) activation function to bound the actions. Uniquely, our objective function diverges from the standard environment reward—which mixes velocity with energy penalties—by explicitly decoupling the control cost; we formulate the objective as the minimization of the negative average forward velocity over a fixed horizon of $H = 1,000$ steps. This objective is subject to a cumulative energy constraint, requiring the total control cost (sum of squared actions, $\sum \|a_t\|^2$) to remain below a threshold of $1,500$. To mitigate the effects of environmental stochasticity, the objective and constraint values are averaged over 5 independent episodes with fixed seeds, and episodes ending prematurely due to instability are penalized via worst-case cost padding for the remainder of the horizon.

## E. Algorithm Setup

We performed all simulations on a server with dual Intel Xeon Platinum 8260 CPUs and 376 GB of memory. To ensure numerical stability during GP training, the design variables $\boldsymbol{x}$ are normalized to the unit interval $[0, 1]^d$ using Min-Max scaling, while the response variables $\boldsymbol{y}$ are standardized to zero mean and unit variance via Z-score transformation. The hyperparameters employed in our algorithm are detailed in Table 1. To ensure a fair comparison, we removed the isometric embedding preprocessing step from HDsafeBO. For all benchmarks, GPs with a RBF kernel serve as the surrogate models for both the objective and constraint functions. Regarding the GP hyperparameters, we utilize fixed ground-truth values for the synthetic benchmarks. Conversely, for the real-world benchmarks, we assign empirical priors to the lengthscale and outputscale to mitigate overfitting, specified as $Gamma(3.0, 3.0)$ and $\mathcal{N}(2.0, 1.0)$, respectively. Furthermore, these hyperparameters are updated via Maximum Likelihood Estimation (MLE) every five iterations. The likelihood noise is set to a constant value of 0.01.

*Table 1.* Algorithm Hyperparameters

| Method | Hyperparameters | Synthetic | | | Truss | Beam | Halfcheetah |
|---|---|---|---|---|---|---|---|
| | | 25 | 50 | 100 | | | |
| CEI | AF_Optimizer | Adam(lr=0.01, num_starts=10) | | | | | |
| EPBO | AF_Optimizer | Adam(lr=0.01, num_starts=10) | | | | | |
| | $\beta$(LCB) | 2.0 | | | | | |
| | penalty_scheduler | $10.0 \times \log(k+1)$ | | | | | |
| SCBO | length_init | 0.8 | | | | | |
| | length_min | $2^{-7}$ | | | | | |
| | length_max | 1.6 | | | | | |
| | perturbation probability | $\min\{1, 20/d\}$ | | | | | |
| | $b$ (batch) | $\lceil d/2 \rceil$ | | | | | |
| | success_tolrance | $\max(3, \lceil d/10 \rceil)$ | | | | | |
| | failure_tolrance | $\lceil d/b \rceil$ | | | | | |
| | n_candidates | 5000 | | | | | |
| HDsafeBO | length_init | 0.8 | | | | | |
| | length_min | $2^{-7}$ | | | | | |
| | length_max | 1.6 | | | | | |
| | $b$ (batch) | $\lceil d/2 \rceil$ | | | | | |
| | success_tolrance | $\max(3, \lceil d/10 \rceil)$ | | | | | |
| | failure_tolrance | $\lceil d/b \rceil$ | | | | | |
| | n_candidates | 5000 | | | | | |
| | $\beta$(LCB) | 2.0 | | | | | |
| LCBO | AF_Optimizer | Adam(lr=0.01, num_starts=10) | | | | | |
| | penalty_scheduler | $10.0 \times k^{1/4}$ | | | | | |
| | $\delta$ | 0.1 | | | | | |
| | $N_m$ | $2d$ | | | | | |
| | $b_t^{(1)}$ | 2 | 2 | 2 | 2 | 1 | 1 |
| | $b_t^{(2)}$ | 5 | 5 | 5 | 5 | 1 | 1 |
| | step_size_scheduler | 0.25 | 0.5 | 0.5 | 0.25 | 0.5 | 0.5 |

