# OpenReview forum: "Local Constrained Bayesian Optimization"
_ICML.cc/2026/Conference — ICML 2026 regular_

### Official Review · Reviewer_cXcn · 2026-03-07

**Soundness:** 3
**Presentation:** 3
**Significance:** 3
**Originality:** 2
**Overall Recommendation:** 4
**Confidence:** 3

**Summary:**

This paper introduces  **Local Constrained Bayesian Optimization (LCBO)**, a framework that directly extends the **GIBO (Local BO)** paradigm from unconstrained policy search to high-dimensional, constrained black-box problems. By leveraging local GP models and a differentiable penalty landscape, LCBO alternates between uncertainty-driven exploration and rapid local descent.

**Compliance With Llm Reviewing Policy:**

Affirmed.

**Key Questions For Authors:**

1. Choice of $\rho$ : Since there is no ablation study for $\rho_k$, how much wiggle room is there for this parameter?  For instance, if a constant penalty were used, would the performance significantly degrade? Do you believe there exist alternative designs for this parameter that would still ensure **Theorem 5.5** holds?

2. Regarding the noise level: In the synthetic experiments, the noise is fixed at $\sigma = 0.1$ (which can be set when generating the data), matching the GP prior. However, in real-world RL tasks where the noise level is unknown, how did you determine the noise hyperparameter? Specifically, did you use repeated sampling at the same point to estimate the noise variance?

**Limitations:**

See the **Weaknesses** section above.

**Strengths And Weaknesses:**

### Strengths
1. The authors successfully adapt the **GIBO framework** to handle complex constraints, filling a gap in high-dimensional constrained Bayesian optimization (CBO).

2. The results on benchmarks and real world data are impressive. LCBO shows better stability and sample efficiency than global baselines.

3. The authors claim a KKT residual convergence rate with a polynomial dependence on dimension $d$. I did not check the proof in detail here.

---

### Weaknesses
1. The reliance on a specific scheduling plan for the penalty parameter $\rho_k$ (e.g., the $k^{1/4}$ growth) feels somewhat prescriptive. In a messy black-box setting, I’m not convinced these theoretical schedules will hold up without significant per-task tuning.

2. I did not find a ablation study regarding the penalty parameter $\rho_k$. While the local BO approach clearly works, it is difficult to determine whether the success stems purely from the local search paradigm from GIBO or if the performance is highly sensitive to this specific penalty tuning.

---

> ### Author Rebuttal · Authors · 2026-03-31
>
> ### W1-2/Q1: On the penalty schedule and its ablation
> We thank the reviewer for this insightful question, which touches on both the theoretical flexibility and practical robustness of our penalty design. We address these concerns from three complementary perspectives.
>
> **Theoretical flexibility.** The $k^{1/4}$ schedule is not the only viable choice. The key requirement on $\rho_k$ in our proof is that the series $\sum_k (\rho_{k+1} - \rho_k)/\rho_k^2$ converges. We selected $\rho_k = k^{1/4}$ specifically because, combined with the corresponding step size $\eta_k$, it ensures that both KKT residuals (excluding the estimation error $\mathcal{E}_K$) achieve the $O(1/\sqrt{K})$ sublinear rate. We note that alternative schedules satisfying the summability condition would necessitate a corresponding adjustment of the step size $\eta_k$, which may alter not only the rate constants but also the specific exponents in the dimensional dependence of the KKT residual bound. Crucially, however, the dependence on $d$ remains polynomial under any such valid schedule, preserving the central theoretical advantage of LCBO over global CBO methods whose regret scales exponentially with $d$.
>
> **Practical robustness across tasks.** To address the concern about per-task tuning, we note that our implementation applies Z-score normalization to all function outputs before GP fitting. This standardization largely removes the influence of problem-specific constraint magnitudes on the effective penalty scale, which is a key reason why a single schedule ($10.0 \times k^{1/4}$) works across all benchmarks in Table 1 without any task-specific adjustment.
>
> **Ablation study: growing schedule vs. fixed constants.** To directly isolate the role of the penalty design within our local search framework, we conducted an ablation comparing LCBO's growing schedule against four fixed-penalty variants ($\rho \in \{1, 10, 100, 1000\}$) on the Truss and 50D Synthetic tasks. Since all variants share the same local exploration-exploitation mechanism, any performance difference is attributable solely to the penalty strategy. The table below reports the median best feasible objective at 1,000 evaluations (lower is better); full convergence curves are provided in https://anonymous.4open.science/r/29738-fig/rho.pdf.
> | Setting | Truss| 50D Synthetic|
> |---|---|---|
> | $\rho = 1$ | 671.7 | -7.503 |
> | $\rho = 10$ | 333.3 | -7.389 |
> | $\rho = 100$ | 332.9 | -7.519 |
> | $\rho = 1000$ | 338.7 | -7.261 |
> | **LCBO** ($10 \times k^{1/4}$) | **323.9** | **-7.514** |
>
> The key finding is that the **optimal fixed constant is task-dependent, and choosing incorrectly incurs a severe penalty**. On the Truss task, $\rho=1$ catastrophically fails (671.7, never improving from initialization), because the penalty is too weak to enforce the tight stress and displacement constraints whose violation dominates the landscape. On the Synthetic task, the ranking reverses: $\rho=1000$ yields the worst result (-7.261), as the excessively large penalty overwhelms the objective signal and impedes progress toward the optimum. No single fixed constant performs well on both tasks simultaneously.
>
> In contrast, LCBO's growing schedule achieves the best or near-best performance on both tasks **without any problem-specific tuning**. This robustness arises from its adaptive nature: moderate penalization in the early phase allows the algorithm to explore and make progress on the objective, while the gradually increasing $\rho_k$ ensures that constraint enforcement tightens as the iterates approach feasibility. In a true black-box setting where the user has no prior knowledge of constraint tightness, this adaptivity eliminates the risk of selecting an inappropriate fixed constant, making the growing schedule a reliable default choice.
> ### Q2: On the noise level in the RL task
> We appreciate the reviewer's attention to this practical detail. As described in Appendix C.3, we mitigate the inherent stochasticity of the MuJoCo simulator by averaging both the objective and constraint values over 5 independent episodes with fixed random seeds. This procedure renders the underlying function nearly deterministic. We then add synthetic Gaussian noise ($\sigma = 0.1$) to match the noise model assumed by all compared methods. The GP noise hyperparameter is accordingly fixed at $\sigma = 0.1$ throughout optimization, rather than estimated online. This choice not only reflects the known noise level but also avoids numerical instability that can arise from MLE-based noise estimation on near-deterministic observations, where the estimated noise variance may collapse to near-zero values and destabilize the GP posterior computation.

---

> > ### Author Rebuttal · Reviewer_cXcn · 2026-04-02
> >
> > I appreciate the authors' clarification regarding the noise setup in the RL tasks. It is helpful to understand that the experiments used controlled synthetic noise ($\sigma=0.1$) to ensure numerical stability. However, my original concern regarding the practical application of LCBO remains: in a true black-box scenario where the noise level is unknown and cannot be manually set, how should a practitioner determine this hyperparameter? I see this might be out of the scope of this paper, though including a brief discussion or a small sensitivity analysis on the noise hyperparameter in the final version would further strengthen the practical value of the work.

---

> > > ### Author Response · Authors · 2026-04-03
> > >
> > > We sincerely thank the reviewer for the follow-up comment and apologize for misinterpreting the original question. The reviewer raises a valid practical concern: how should a practitioner determine the noise hyperparameter in a true black-box setting where the noise level is unknown? We address this with new experiments on the HalfCheetah task (our most challenging benchmark with $d=102$).
> > >
> > > **MLE-based noise estimation.** In standard BO practice, GP hyperparameters (including the noise variance) are routinely optimized via Maximum Likelihood Estimation (MLE). To verify that this approach is sufficient for LCBO, we treated the noise variance as a free parameter optimized jointly with other kernel hyperparameters. The results show that LCBO with MLE-estimated noise achieves performance comparable to the fixed-noise setting, confirming that practitioners can rely on MLE without manually specifying the noise level, following the same protocol commonly used for lengthscale and outputscale.
> > >
> > > **Sensitivity to fixed noise values.** We further evaluated LCBO under misspecified noise variances ($\sigma^2 \in \{0.005, 0.05\}$, compared to the original $\sigma^2 = 0.01$). All four configurations (three fixed values and MLE) exhibit similar optimization trajectories, demonstrating that LCBO is robust to the noise hyperparameter choice. This is consistent with the broader GP literature, where a moderate noise term often serves as a numerical jitter to stabilize posterior computation, and exact calibration is not critical for optimization performance.
> > >
> > > We note that all baselines in our main experiments share the same noise configuration, so the comparative conclusions remain unaffected. We will include this sensitivity analysis and a brief discussion on noise hyperparameter selection in the camera-ready version. The detailed figure is available at https://anonymous.4open.science/r/29738-fig/noise.pdf.

---

### Official Review · Reviewer_7HmP · 2026-03-09

**Soundness:** 4
**Presentation:** 3
**Significance:** 4
**Originality:** 2
**Overall Recommendation:** 5
**Confidence:** 3

**Summary:**

This paper addresses the problem of high-dimensional constrained black-box optimization (CBO), i.e., the optimization of problems that can only be observed point-wise and that have unknown constraints. Such problems occur across multiple applications, for instance, in drug discovery, where a candidate must be synthesized before its properties can be measured.

Motivated by the need for high-dimensional CBO methods that do not rely on trust regions, the authors propose an extension of the GIBO method to constrained problems. The proposed method, local constrained Bayesian optimization (LCBO), models the constrained problem with quadratic penalties and alternates between local exploration and local exploitation steps. A local exploration step aims to reduce uncertainty about the "current" solution $x_k$ by selecting the point of maximum gradient uncertainty (leveraging the fact that with a Gaussian process (GP) surrogate, the posterior of the gradient of the surrogate also is a GP), while the local exploitation phase updates $x_k$ by following the gradient of the surrogate model.

The authors establish a polynomial convergence bound for the proposed algorithm and benchmark LCBO on six constrained problems, demonstrating that it outperforms the state of the art on all of them.

**Compliance With Llm Reviewing Policy:**

Affirmed.

**Final Justification:**

The rebuttal addressed my concerns, and I have already raised my initial score.

**Key Questions For Authors:**

- What are the reasons why LCBO outperforms SCBO? Have you observed some particular failure modes of SCBO that LCBO solves?
- What is the impact when further tightening the constraints (as described in line 333, p.7)
- What is the most important learning from your theoretical contributions?

**Limitations:**

yes

**Strengths And Weaknesses:**

**Strengths**:

- The paper is well-written. The notation is clear, and the theoretical analysis seems rigorous (without reading the proofs in the appendix).
- The motivation for the method is clear, and the overall narration is easy to follow.
- The paper addresses a relevant problem that occurs in various disciplines, including important applications in engineering and life sciences.
- The method performs well against a set of reasonably chosen baselines. Furthermore, the empirical evaluation is rigorous, so that LCBO is a promising new state of the art for constrained high-dimensional black-box optimization.

**Weaknesses**:

- The paper claims that LCBO is better suited for handling problems where the primary descent direction is obstructed by complex or tight constraints. While the authors show empirically that LCBO outperforms the state of the art, I'm missing additional arguments for why LCBO should be better at dealing with those scenarios.
- The novelty of the method is limited. The method is an extension of GIBO (a fact the authors are honest about) to constrained problems.
- GIBO itself is only briefly explained, although it sets the foundation for LCBO. It would be easier to understand this paper's contribution if GIBO were explained in more detail so that the paper could indicate which parts of the method are new.
- Some of the definitions read like they are contributions by the authors (e.g., Defs 3.1 - 3.3), although at least some of them define preexisting concepts, which should be cited.

---

> ### Author Rebuttal · Authors · 2026-03-31
>
> ### W1/Q1: Why LCBO Is Better Suited for Tight/Complex Constraints
> The core failure mode of SCBO arises when the trust-region center approaches a constraint boundary that curves or narrows: candidates improving the objective tend to be infeasible, while feasible candidates offer no improvement. SCBO's radius update, governed solely by whether a better feasible point is found, then inevitably triggers shrinkage and restarts, consuming substantial evaluation budget without meaningful optimization progress. LCBO sidesteps this entirely by updating via gradient descent on the penalized surrogate, where the penalty term naturally steers iterates along the constraint boundary without requiring any geometric region management. We provide a 2D illustrative diagram at https://anonymous.4open.science/r/29738-fig/mechanism.pdf depicting this contrast, and empirical evidence (SCBO restart counts on the truss task) in our response to Reviewer PSVN, Q5.
> ### Q2: Tighter Constraints
> We conducted an ablation on the 50D synthetic benchmark by varying the constraint offset (larger offset = narrower feasible region). Results at 1000 evaluations (median of 10 runs):
> | Offset | LCBO | SCBO | Gap |
> |--------|--------|--------|-------|
> | 0.5 | -7.514 | -6.637 | 0.877 |
> | 1.0 | -7.129 | -6.225 | 0.904 |
> | 2.0 | -6.972 | -5.677 | 1.295 |
>
> LCBO's advantage grows consistently as constraints tighten (gap: 0.877→1.295), confirming that gradient-based navigation along constraint boundaries becomes increasingly valuable as the feasible region narrows. Full curves are available at https://anonymous.4open.science/r/29738-fig/offset.pdf.
> ### Q3: Theoretical Contributions
> We establish the **first convergence analysis for local constrained BO**. Corollary 5.6 shows that LCBO's KKT residual scales **polynomially** with $d$ for RBF and Matérn 2.5 kernels, in contrast to the exponential dependence typical of global CBO bounds (e.g., EPBO). This provides the first theoretical justification that local search is a principled, scalable paradigm for high-dimensional CBO. We also refer the reviewer to the detailed technical novelties discussed in our response to W2.
> ### W2: Novelty
> We respectfully argue that the contribution goes well beyond a straightforward extension of GIBO to the constrained setting. While the high-level idea of combining local search with a penalty function may appear natural, we demonstrate that a naive implementation fails, and our specific design choices are critical.
>
> **Naive version fails.** We implemented "NaiveLCBO", which models the penalized objective $Q_\rho(x)$ with a single GP and applies GIBO directly, selecting the best $\rho$ from $\{1, 10, 100\}$ (giving it an oracle-level hyperparameter advantage). As shown in https://anonymous.4open.science/r/29738-fig/naive.pdf, NaiveLCBO performs dramatically worse than LCBO on both the Truss and 50D Synthetic tasks, demonstrating that the seemingly natural approach is fundamentally inadequate.
>
> **Non-trivial technical contributions.** The key designs distinguishing LCBO include:
>
> (1) *Minimax acquisition function*: Separate GPs for the objective and each constraint, with a minimax acquisition function (Eq. 12–13) that aggregates gradient uncertainty across all GPs. This design not only yields strong empirical performance but is also essential for establishing the theoretical convergence bounds.
>
> (2) *Single-loop penalty method*: Classical penalty methods require solving an inner subproblem to convergence per $\rho$, which is infeasible under noisy black-box access. Our algorithm performs one gradient step per iteration with $\rho_k = k^{1/4}$, which we prove sufficient for convergence (Theorem 5.5).
>
> (3) *Tractable gradient approximation*: The exact posterior of $\nabla c(x)^\top c(x)$ is non-Gaussian due to correlated GP quantities (Remark 4.1). Our decomposed approximation (Eq. 14) remains Gaussian and amenable to tight high-probability bounds.
>
> (4) *Theoretical analysis*: Extending convergence analysis to the constrained setting requires handling the interplay between penalty scheduling, the regularity condition (Assumption 5.3), and cumulative estimation error across multiple GPs, which is a substantial contribution in its own right.
> ### W3: Explanation of GIBO
> In the revision, we will expand the description of GIBO to clearly delineate inherited vs. new components. Specifically, we will summarize GIBO's two-phase (exploration-exploitation) mechanism for unconstrained problems, then explicitly highlight LCBO's modifications: the  minimax acquisition function (Eq. 12–13) handling multi-GPs, the penalized gradient update, and the single-loop penalty scheduling.
> ### W4: Citations for Definitions
> Definitions 3.1–3.3 are standard or adapted from prior work, not original contributions. In the revision, we will add appropriate citations: L-smoothness (Nesterov, 2018), KKT residuals (Nocedal, 2006), and the error function adapted from Wu et al. (2023) to our constrained setting.

---

> > ### Author Rebuttal · Reviewer_7HmP · 2026-04-01
> >
> > The rebuttal addressed all my concerns.

---

> > > ### Author Response · Authors · 2026-04-02
> > >
> > > We sincerely thank the reviewer for the constructive feedback and for re-evaluating our work. We are glad that our response addressed the concerns. We will incorporate the suggestions into the final version of the paper.

---

### Official Review · Reviewer_PSVN · 2026-03-11

**Soundness:** 4
**Presentation:** 3
**Significance:** 3
**Originality:** 3
**Overall Recommendation:** 5
**Confidence:** 4

**Summary:**

This paper proposes Local Constrained Bayesian Optimization (LCBO), a method for optimizing problems with expensive black-box objectives and constraints in high-dimensional spaces, where traditional Bayesian optimization struggles due to the curse of dimensionality. LCBO replaces rigid trust-region strategies with a constraint-penalized surrogate model, allowing the algorithm to alternate between local descent steps and uncertainty-driven exploration to better navigate complex feasible regions. The authors provide theoretical results showing that the KKT residual converges at a rate that scales polynomially with the problem dimension for common kernels. Experiments on benchmark problems up to 100 dimensions show that LCBO achieves strong sample efficiency and outperforms existing constrained Bayesian optimization methods.

**Compliance With Llm Reviewing Policy:**

Affirmed.

**Final Justification:**

LGTM. Thank you for the thorough response.

**Key Questions For Authors:**

See above for questions.

**Limitations:**

Yes

**Strengths And Weaknesses:**

The paper is sound. Its claims are well-supported by theoretical analysis and numerical experiments. Presentation raises some questions:
- Algorithm 1 says, "Solve for active learning candidates". This is the only use of the phrase "active learning" in the paper. The phrase should be explained.
- No constraint for the HalfCheetah problem is mentioned in the main text. I believe Appendix C describes it. Since this problem is the only truly high-dimensional, real-world example in the paper, its constraint should be made explicit in the main text. Readers may misread it as being unconstrained, especially since HalfCheetah is normally presented without a constraint.
- employing a fixed batch size in each iteration to guarantee sample efficiency." How does this *guarantee* sample efficiency? How does performance vary with batch size?
- It seems to be an important point that LCBO does not use a trust region. You say, "Unlike trust-region methods that rely on rigid geometric regions and are prone to premature shrinkage, LCBO...". Yet, you later say that for algorithmic efficiency, you are "restricting the search space for optimizing the acquisition function Eq. (12) to the dynamic local region [xk−δ,xk+δ]"   You've specified bounds on x. Isn't that a trust region?
- The abstract and introduction mention "premature shrinkage" of trust region methods. The intro says, "we explicitly demonstrate on the truss design task", and in section 6.3, "as their regions shrink against the constraints. ".  Yet, you don't report the trust region size. Does it actually shrink? Does SCBO stagnate *because* the trust region shrinks?
- "maintaining a local GP model trained exclusively on the N_m most recent data points"  How does optimization performance vary with N_m? Presumably you introduce N_m b/c of the poor scaling of GPs with number of observations, but how much optimization performance do you need to sacrifice in order get the optimization to complete in the amount of time you have allotted?

---

> ### Author Rebuttal · Authors · 2026-03-31
>
> ### Q1: Terminology in Algorithm 1
> We agree that "active learning" carries a distinct connotation in the broader ML literature and may mislead readers. We will replace it with **"active exploration candidates"** throughout to align with our exploration–exploitation framework.
> ### Q2: Constraint definition
> The reviewer is correct that standard HalfCheetah is typically unconstrained, and omitting the constraint from the main text risks misinterpretation. We will add to Section 6.3: the objective minimizes negative average forward velocity over H=1,000 steps, subject to a cumulative energy constraint requiring $\sum\|a_t\|^2 \leq1{,}500$.
> ### Q3: Batch size and sample efficiency
> We agree the phrase "guarantee" is imprecise. The intended argument is that under a fixed budget, smaller batches permit far more frequent updates (e.g., \~700 vs. <50 iterations in 100D with 5,000 evaluations), yielding better anytime performance despite noisier per-step estimates. Our ablation in Appendix B (Figures 3–4) confirms that while larger batches achieve steeper per-iteration descent, the mini-batch strategy delivers superior per-evaluation performance. We will revise the phrasing to: *"a fixed mini-batch size to improve sample efficiency"*
> ### Q5: Discussion for premature shrinkage in SCBO
> We thank the reviewer for this important question. We acknowledge that our original phrasing did not explicitly delineate the causal mechanism; we will revise the manuscript accordingly.
>
> **Mechanism analysis.** We provide an illustrative diagram (https://anonymous.4open.science/r/29738-fig/mechanism.pdf) comparing SCBO and LCBO on a constrained landscape where the optimum lies on a curved constraint boundary. When SCBO's trust region center approaches such a boundary, the candidates face a structural dilemma: points along the descent direction violate the constraints, while feasible points within the region offer no improvement. Both outcomes register as failures under SCBO's update rule, triggering radius reduction. Shrinkage is a consequence of this geometric conflict, but once triggered it creates a compounding effect: a smaller region overly restricts exploration, making it increasingly difficult to escape the current neighborhood. Our original text may give the impression that shrinkage directly causes stagnation, rather than being an amplifying consequence of the geometric conflict. We will revise the introduction and Section 6.3. By contrast, LCBO uses gradient information from the penalized surrogate to navigate along the constraint boundary directly.
>
> **Empirical confirmation on the truss task.** In SCBO, a restart is triggered if and only if the radius falls below `length_min`, constituting direct evidence of shrinkage. Across all 10 runs, every run experienced at least 2 restarts, with the first at median 350 evaluations (IQR: 338–383) and the second at 747 (IQR: 657–756). Notably, the timing of the first restart aligns closely with the plateau onset in Figure 2(a), providing direct empirical confirmation that the geometric obstruction described above is responsible for SCBO's performance plateau on this task. We acknowledge that restarts can in principle allow recovery, and the evaluations during shrinkage phases are not entirely wasted since they still contribute to the GP model. However, these phases tend to allocate a substantial portion of the budget to candidates that yield no feasible improvement.
> ### Q4 & Q6: Local search box and local GP
> **The local box is not a trust region.** The key distinction is twofold. First, the box [x_k−δ, x_k+δ] only constrains the exploration phase (optimization of Eq. 12, determining where to collect observations); the exploitation step (Line 11), where xk is actually updated, is constrained solely by the original domain X. In SCBO, the trust region constrains both. Second, δ is fixed throughout optimization with no adaptive shrinkage/expansion mechanism.
>
> **Motivation: local gradient information.** Both the local box and the local GP (N_m=2d, Table 1) are motivated by the same principle: since LCBO performs gradient-based local search, optimization decisions at x_k depend primarily on the nearby landscape. The local box focuses exploration where observations are most informative for gradient estimation at x_k. The local GP, by training only on recent data, ensures the surrogate reflects the local landscape rather than being diluted by distant, less relevant observations. This also provides implicit adaptability to non-stationary processes (Muller et al., 2021) and reduces per-iteration cost from O(N³) to O(N_m³).
>
> **Ablation results.** Our ablation study (https://anonymous.4open.science/r/29738-fig/local.pdf) shows that both components actually *improve* performance compared to their removal: disabling the local box or using a global GP both degrade optimization quality on the truss and 50D synthetic tasks. These are design choices aligned with the local nature of our method.

---

> > ### Author Rebuttal · Reviewer_PSVN · 2026-04-02
> >
> > > "active exploration candidates"
> >
> > You still need to define clearly how that is done and connect the mention of it in the algo box to any discussion in the paper (e.g., with consistent terminology or an explicit reference).
> >
> > > "a fixed mini-batch size to improve sample efficiency"
> >
> > This seems too imprecise. If you have a claim that is justified by the appendix, you should state the claim clearly and reference the appendix.
> >
> > > Our ablation study (https://anonymous.4open.science/r/29738-fig/local.pdf) shows
> >
> > These plots show very little information: (i) LocalGP looks to have no impact, (ii) Given the size of the errorbars and small size of the performance difference, the impact of LocalBox might very well disappear when the studies are run over more tasks.

---

> > > ### Author Response · Authors · 2026-04-03
> > >
> > > We thank the reviewer for the careful follow-up. The points raised are well-taken, and we address each below with concrete revisions.
> > > ### Active exploration candidates
> > >
> > > We appreciate the reviewer pointing out the disconnect between Algorithm 1 and the main text. The issue is not merely the phrase itself, but that Algorithm 1 Line 6 uses terminology ("active learning candidates") that appears nowhere else in the paper, while the corresponding discussion (Section 4, "Local Exploration" paragraph) describes the same operation without referencing the algorithm box. We will
> > >
> > > (1) replace Line 6 with "Select exploration batch $Z_2$​ by minimizing the acquisition function (Eq. 12)"
> > >
> > > (2) In the Local Exploration paragraph (Section 4), add an explicit forward reference: "...we select a batch of exploration candidate points $Z_2$ (Line 6 of Algorithm 1) by minimizing..."
> > >
> > > This ensures bidirectional consistency between the algorithm box and the prose.
> > > ### Fixed mini-batch
> > > We agree the main text should state the claim precisely and reference the appendix. We will revise the sentence to:
> > >
> > > *"We employ a fixed mini-batch size at each iteration. Although our theoretical analysis assumes growing batch sizes for exact convergence (corollary 5.6), under finite evaluation budgets, fixed mini-batches yield superior per-evaluation performance by enabling substantially more frequent updates (e.g., \~700 vs. fewer than 50 iterations in the 100D setting with 5,000 evaluations). A systematic ablation comparing fixed, large-batch, and growing schedules is provided in Appendix B (Figures 3–4)."*
> > > ### Local GP and Local Box
> > > We believe the reviewer's observations on the ablation plots are accurate, and we would like to reframe our argument accordingly, as our original rebuttal may have conveyed a stronger claim than the evidence supports.
> > >
> > > **Local GP.** The reviewer correctly observes that the local GP has negligible impact on optimization performance. This observation, in fact, directly answers the reviewer's original Q6: *the performance sacrifice from restricting the GP to the $N_m$ most recent points is minimal*. The primary motivation for the local GP is computational: reducing per-iteration cost from $O(N^3)$ to $O(N_m^3)$, a standard practice inherited from GIBO (Müller et al., 2021). The ablation confirms that this computational shortcut does not degrade optimization quality, which is exactly the property we need.
> > >
> > > **Local Box.** We acknowledge the reviewer's concern that the observed improvement may not generalize across a broader task suite given the error bars. Rather than claiming performance gains, we clarify the role of the local box as follows:
> > >
> > > 1. Its primary function is computational: constraining the acquisition function optimization (Eq. 12) to a local neighborhood avoids costly global search over the full high-dimensional domain.
> > >
> > > 2. It is consistent with the local search philosophy: since the acquisition function (Eq. 12–13) measures gradient uncertainty *at $x_k$*, observations far from $x_k$ contribute diminishing information, making local search a natural and sufficient strategy.
> > >
> > > 3. As stated in our response to Q4, the local box differs structurally from a trust region: it constrains only the exploration phase (where to collect observations), not the exploitation step (how $x_k$ is updated), and its size $\delta$ remains fixed throughout optimization.
> > >
> > > **Reframing the ablation.** To be precise about what the ablation demonstrates: the performance advantage of LCBO over baselines (Figures 1–2) is attributable to the core algorithmic framework (penalty-based local search with gradient descent on the penalized surrogate), not to the local GP or local box individually. The ablation confirms that these implementation choices, motivated by computational efficiency and the local nature of the method, do not harm and may modestly help optimization performance. We will revise the discussion in the paper to reflect this framing.

---

### Official Review · Reviewer_YHxa · 2026-03-14

**Soundness:** 2
**Presentation:** 3
**Significance:** 2
**Originality:** 3
**Overall Recommendation:** 4
**Confidence:** 4

**Summary:**

This paper studies high dimensional constrained Bayesian optimization and proposes LCBO, a local method that alternates a local exploration step that reduces GP gradient uncertainty near the current point and a local exploitation step that descends a quadratic penalty surrogate. The paper gives a convergence analysis in terms of average KKT and feasibility residuals with polynomial dependence on dimension for RBF and Matern kernels under regularity assumptions. Experiments on synthetic tasks up to 100D and on truss, beam, and HalfCheetah tasks show better best feasible objective than CEI, EPBO, SCBO, and HDsafeBO under the reported setup.

**Compliance With Llm Reviewing Policy:**

Affirmed.

**Final Justification:**

The authors' answers have resolved my concerns.  Thus, I increase my score as weak accept.

**Key Questions For Authors:**

- Which parts of the empirical implementation are essential for the gains beyond the fixed mini batch change, especially the local GP window, the local search box, and gradient normalization?
- Can the authors report feasibility focused metrics such as time to first feasible point and median constraint violation over time for the main tasks?
- How much do the results change if HDsafeBO is run with its full preprocessing pipeline and if a direct local penalty baseline is added?

**Limitations:**

Yes

**Strengths And Weaknesses:**

## Strengths

- The problem is important, and the paper targets a real gap between low dimensional constrained BO and high dimensional local BO.
- The algorithmic idea is clear and reasonably simple. The alternating local uncertainty reduction and penalty descent steps are easy to follow and seem implementable in existing GP based BO pipelines.
- The theory is stronger than a purely empirical paper. The KKT residual result with polynomial dependence on $d$ is a meaningful contrast with the exponential dimension dependence discussed for global constrained BO.
- The empirical section covers both synthetic and application style tasks, including a 102D control problem, and the curves suggest consistent gains in sample efficiency.

## Weaknesses

- The biggest issue is the gap between the proved algorithm and the implemented one. The theorem assumes growing batch sizes and the analysis is developed for equality constraints, while the experiments use fixed mini batches, local GPs over recent points, gradient normalization, local search box restriction, smooth max approximations, RFF sampling, and inequality style engineering tasks. The appendix only studies the batch schedule, so the theoretical claim does not fully justify the actual method used in the experiments.
- The empirical evaluation reports only best feasible objective. For a constrained method, feasibility rate, violation magnitude, time to first feasible point, and possibly wall clock cost would be important. Several claims about handling tight constraints and avoiding premature shrinkage would be more convincing with direct constraint metrics.
- The baseline comparison could be stronger. The paper removes the isometric embedding step from HDsafeBO, but does not show that this change preserves a competitive baseline. It would also help to compare against a more direct local constrained adaptation of GIBO or another penalty based local BO baseline, since the proposed method is conceptually closest to that family.
- The constraint modeling story is somewhat unclear. The main formulation is equality constrained and suggests handling inequalities through slack variables, but the experiments seem to use aggregated inequality constraints directly. That leaves some ambiguity about what problem class LCBO is actually designed and analyzed for.

---

> ### Author Rebuttal · Authors · 2026-03-31
>
> ### W1/Q1: Theory–practice gap
> We first clarify three items that are not gaps: (i) RFF is used only to construct synthetic ground-truth functions from a GP prior, not in our algorithm; (ii) inequality constraints are addressed in W4; (iii) the LogSumExp smooth-max is a standard technique for enabling gradient-based acquisition optimization. We now address the genuine gaps:
>
> **Fixed mini-batches.** Theory requires growing batches; in practice, fixed mini-batches yield better sample efficiency under finite budgets. The ablation in Appendix B confirms this across 25–100D.
>
> **Local GP, local search box, and gradient normalization.** All three originate from GIBO: local GPs are trained on the N_m most recent points, reducing training cost from O(N³) to O(N_m³) while adapting to local landscape changes. local box is used for informative points near the current iterate. Gradient normalization decouples direction from magnitude, providing adaptive step-size control akin to Adam/RMSprop. We ablated each component individually on the Truss and 50D Synthetic tasks. All ablated variants remain competitive with or outperform the strongest baseline, confirming that the core mechanism, penalty-based gradient descent on GP surrogates, drives the primary gains. Full ablation curves are available at https://anonymous.4open.science/r/29738-fig/wo_ablation.pdf.
> ### W2/Q2: Feasibility and wall-clock cost
> We report feasible rate, median constraint violation, and wall-clock time at https://anonymous.4open.science/r/29738-fig/tables.pdf. Regarding time to first feasible point: our cold-start protocol uses d Sobol points for initialization, and infeasible values are treated as infinity in Figures 1–2. Thus, the onset of each curve already reflects when feasibility is first achieved.
>
> For **feasibility metrics**, LCBO achieves substantially lower constraint violation than global methods (CEI, EPBO) across tasks, indicating its infeasible evaluations concentrate near the constraint boundary. LCBO's feasible rate is competitive overall and notably the highest on the Truss task. We note that in CBO literature, best feasible objective is the standard primary metric (Eriksson & Poloczek, 2021), as the goal is finding the best feasible solution rather than minimizing violations during search (Remark 1.1).
>
> For **wall-clock time**, LCBO (\~1.37 s/eval on 50D synthetic task with 1000 evaluations) is slower per evaluation than trust-region methods (\~0.05–0.07 s) but faster than global GP methods (CEI ~2.41 s, EPBO ~2.08 s). Unlike global methods whose cost grows with dataset size, LCBO's local GP yields constant per-evaluation cost. More importantly, BO targets expensive-to-evaluate functions where the evaluation cost t\_eval dominates. On the 50D synthetic task, LCBO reaches the objective value that SCBO attains at 500 evaluations using fewer than half the evaluations. For a moderate t\_eval = 60 s (typical for FEA simulations), this sample-efficiency advantage translates to roughly 2× reduction in total wall-clock time despite the higher per-evaluation overhead.
>
> Regarding **claims about tight constraints and premature shrinkage**, we refer to Reviewer 7HmP (Q2), where we ablate constraint tightness on Synthetic-50D, and to Reviewer PSVN (Q5), where we analyze the distinct optimization mechanisms of SCBO vs. LCBO.
> ### W3/Q3: Baseline comparisons
> The embedding module and the trust-region search in HDsafeBO are largely independent components; embedding could equally be prepended to SCBO, CEI, or EPBO. Retaining it only for HDsafeBO would conflate dimensionality-reduction benefits with those of its trust-region mechanism. Moreover, the original HDsafeBO experiments target a very different regime (4000D→50D, compression ratio 80:1), where the sparse subspace assumption is considerably less justified in our 25–100D setting. An ablation comparing HDsafeBO with and without PCA embedding (mapping to 10D) shows consistent degradation when embedding is applied, supporting the fairness of our setup. Full results are at https://anonymous.4open.science/r/29738-fig/hd_full.pdf.
>
> On the local penalty baseline: to our knowledge, no existing method combines GP-based gradient estimation with a penalty formulation in a local framework, so a direct "local penalty" comparator does not exist. To address this concern, we constructed a naive local-penalty baseline for head-to-head comparison; results confirm LCBO's advantage (see Reviewer 7HmP, W2).
> ### W4: Constraint formulation
> The equality formulation in Eq. (1) is chosen for generality, as inequalities reduce to equalities via slack variables (Picheny et al., 2016). In implementation, inequality constraints are handled via the standard one-sided penalty, preserving the algorithmic mechanism (Nocedal & Wright, 2006). The LogSumExp aggregation in truss/beam benchmarks consolidates same-type constraints and is applied identically across all baselines. We will clarify both points in the revision.

---

> > ### Author Rebuttal · Reviewer_YHxa · 2026-03-31
> >
> > The authors' answers have resolved my concerns.

---

> > > ### Author Response · Authors · 2026-04-02
> > >
> > > We greatly appreciate the reviewer's thoughtful and detailed evaluation. We are encouraged that our rebuttal has adequately addressed the raised concerns. The reviewer's feedback has been valuable in strengthening our work, and we will ensure all suggested revisions are reflected in the camera-ready version.

---

### Decision · Program_Chairs · 2026-04-30

**Decision:**

Accept (regular)

**Comment:**

This paper considers the setting of Bayes opt in high dimensional and constrained settings. The paper proposes a local constrained Bayes opt approach, and provide theoretical guarantees for its convergence. Finally, the method is demonstrated empirically on high-dimensional benchmarks.

During the rebuttal, the authors provided new results that have strengthened the reviewers' view on this paper. All reviewers currently are unanimous in accepting the paper.